# Implicit Regularization Paths of Weighted Neural Representations

**Jin-Hong Du**
Carnegie Mellon University
jinhongd@andrew.cmu.edu

**Pratik Patil**
University of California Berkeley
pratikpatil@berkeley.edu

## Abstract

We study the implicit regularization effects induced by (observation) weighting of pretrained features. For weight and feature matrices of bounded operator norms that are infinitesimally free with respect to (normalized) trace functionals, we derive equivalence paths connecting different weighting matrices and ridge regularization levels. Specifically, we show that ridge estimators trained on weighted features along the same path are asymptotically equivalent when evaluated against test vectors of bounded norms. These paths can be interpreted as matching the effective degrees of freedom of ridge estimators fitted with weighted features. For the special case of subsampling without replacement, our results apply to independently sampled random features and kernel features and confirm recent conjectures (Conjectures 7 and 8) of the authors on the existence of such paths in [50]. We also present an additive risk decomposition for ensembles of weighted estimators and show that the risks are equivalent along the paths when the ensemble size goes to infinity. As a practical consequence of the path equivalences, we develop an efficient cross-validation method for tuning and apply it to subsampled pretrained representations across several models (e.g., ResNet-50) and datasets (e.g., CIFAR-100).

## 1 Introduction

In recent years, neural networks have become state-of-the-art models for tasks in computer vision and natural language processing by learning rich representations from large datasets. Pretrained neural networks, such as ResNet, which are trained on massive datasets like ImageNet, serve as valuable resources for new, smaller datasets [32]. These pretrained models reduce computational burden and generalize well in tasks such as image classification and object detection due to their rich feature space [32, 69]. Furthermore, pretrained features or neural embeddings, such as the neural tangent kernel, extracted from these models, serve as valuable representations of diverse data [33, 66].

However, despite their usefulness, fitting models based on pretrained features on large datasets can be challenging due to computational and memory constraints. When dealing with high-dimensional pretrained features and large sample sizes, direct application of even simple linear regression may be computationally infeasible or memory-prohibitive [23, 44]. To address this issue, subsampling has emerged as a practical solution that reduces the dataset size, thereby alleviating the computational and memory burden. Subsampling involves creating smaller datasets by randomly selecting a subset of the original data points. Beyond these computational and memory advantages, subagging can also greatly improve predictive performance in overparameterized regimes, especially near model interpolation thresholds [53]. Moreover, through distributed learning, models fitted on multiple subsampled datasets can be aggregated as an ensemble to provide more stable predictions [20, 21, 51].

There has been growing interest in understanding the effects of subsampling (without replacement) [16, 25, 37, 50, 51]. These works relate subsampling to explicit ridge regularization, assuming either

38th Conference on Neural Information Processing Systems (NeurIPS 2024).

Table 1: Overview of related work on the equivalence of implicit regularization and explicit ridge regularization.

| Main analysis | Feature structure | Weight structure | Reference |
|---|---|---|---|
| Risk characterization | Gaussian | subsampling | [37] |
| | linear | subsampling | [51] |
| | Gaussian | bootstrapping | [5, 16, 17] |
| Estimator equivalence | linear | subsampling | [50] |
| | general | general | Theorem 1 |
| | general | subsampling | Theorem 2 |
| | linear, random, kernel | subsampling | Propositions 3–5 |
| Risk equivalence | linear | subsampling | [50] |
| | general | general | Theorem 6 |

Gaussian features $\phi \sim \mathcal{N}(\mathbf{0}_p, \mathbf{\Sigma})$ or linearly decomposable features (referred to as *linear features* in this paper) $\phi = \mathbf{\Sigma}^{1/2}\mathbf{z}$, where $\mathbf{\Sigma} \in \mathbb{R}^{p \times p}$ is the covariance matrix and $\mathbf{z} \in \mathbb{R}^p$ contains i.i.d. entries with zero means and bounded $4 + \delta$ moments for some $\delta > 0$. Specifically, [50] establish a connection between implicit regularization induced by subsampling and explicit ridge regularization through a *path* defined by the tuple $(k/n, \lambda)$, where $k$ and $n$ are the subsample size and the full sample size, respectively, and $\lambda$ is the ridge regularization level. Along this path, any subsample estimator with the corresponding ridge regularization exhibits the same first-order (or estimator equivalence) and second-order (or risk equivalence) asymptotic limits. Moreover, the endpoints of all such paths along the two axes of $k = n$ (no subsampling) and $\lambda = 0$ (no regularization) span the same range. Although these results have been demonstrated for linear features, [50] also numerically observe similar equivalence behavior in more realistic contexts and propose conjectures for random features and kernel features based on heuristic "universality" justifications. However, extending these results to encompass more general feature structures and other sampling schemes remains an open question.

Towards answering this question, in this paper, we view subsampling as a weighted regression problem [67]. This perspective allows us to study the equivalence in its most general form, considering arbitrary feature structures and weight structures. The general weight matrix approach used in this study encompasses various applications, including subsampling, bootstrapping, variance-adaptive weighting, survey, and importance weighting, among others. By interpreting subsampling as a weighted regression problem, we leverage recent tools from free probability theory, which have been developed to analyze feature sketching [39, 42, 54]. Building on these theoretical tools, we establish implicit regularization paths for general weighting and feature structures. We summarize our main results below and provide an overview of our results in the context of recent related work in Table 1.

## 1.1 Summary of results and paper outline

We summarize our main results and provide an outline for the paper below.

- *Paths of weighted representations.* In Section 3, we demonstrate that general weighted models exhibit first-order equivalence along a path (Theorem 1) when the weight matrices are asymptotically independent of the data matrices. This path of equivalence can be computed directly from the data using the formula provided in Equation (2). Furthermore, we provide a novel interpretation of this path in terms of matching effective degrees of freedom of models along the path for general feature structures when the weights correspond to those arising from subsampling (Theorem 2).

- *Paths of subsampled representations.* We further specialize our general result in Theorem 2 for the weights induced by subsampling without replacement to structured features in Section 3.2. These include results for linear random features, nonlinear random features, and kernel features, as shown in Propositions 3–5, respectively. The latter two results also resolve Conjectures 7 and 8 raised by [50] regarding subsampling regularization paths for random and kernel features, respectively.

- *Risk equivalences and tuning.* In Section 4, we demonstrate that an ensemble of weighted models has general quadratic risk equivalence on the path, with an error term that decreases inversely as $1/M$ as the number of ensemble size $M$ increases (Theorem 6). The risk equivalence holds for both in-distribution and out-of-distribution settings. For subsampling general features, we derive an upper bound for the optimal subsample size (Proposition 7) and propose a cross-validation method to tune the subsample and ensemble sizes (Algorithm 1), validated on real datasets in Section 4.3.

This level of generality is achievable because we do not analyze the risk of either the full model or the weighted models in isolation. Instead, we relate these two sets of models, allowing us to maintain weak assumptions about the features. The key assumption underlying our results is the asymptotic freeness of weight matrices with respect to the data matrices. While directly testing this assumption is generally challenging, we verify its validity through its consequences on real datasets in Section 4.3.

## 1.2   Related literature

We provide a brief account of other related work below to place our work in a better context.

*Linear features.* Despite being overparameterized, neural networks generalize well in practice [70, 71]. Recent work has used high-dimensional "linearized" networks to investigate the various phenomena that arise in deep learning, such as double descent [12, 46, 48], benign overfitting [10, 35, 45], and scaling laws [7, 19, 66]. This literature analyzes linear regression using statistical physics [14, 60] and random matrix theory [22, 30]. Risk approximations hold under random matrix theory assumptions [6, 30, 66] in theory and apply empirically on a variety of natural data distributions [43, 60, 66].

*Random and kernel features.* Random feature regression, initially introduced in [56] as a way to scale kernel methods, has recently been used for theoretical analysis of neural networks and trends of double descent in deep networks [1, 46]. The generalization of kernel ridge regression has been studied in [11, 40, 57]. The risks of kernel ridge regression are also analyzed in [9, 19, 29]. The neural representations we study are motivated by the neural tangent kernel (NTK) and related theoretical work on ultra-wide neural networks and their relationships to NTKs [34, 68].

*Resampling analysis.* Resampling and weighted models are popular in distributed learning to provide more stable predictions and handle large datasets [20, 21, 51]. Historically, for ridge ensembles, [36, 61] derived risk asymptotics under Gaussian features. Recently, there has been growing interest in analyzing the effect of subsampling in high-dimensional settings. [37] considered least squares ensembles obtained by subsampling, where the final subsampled dataset has more observations than the number of features. For linear models in the underparameterized regime, [59] also provide certain equivalences between subsampling and iterative least squares approaches. The asymptotic risk characterization for general data models has been derived by [51]. [25, 50] extended the scope of these results by characterizing risk equivalences for both optimal and suboptimal risks and for arbitrary feature covariance and signal structures. Very recently, different resampling strategies for high-dimensional supervised regression tasks have been analyzed by [17] under isotropic Gaussian features. Cross-validation methods for tuning the ensemble of ridge estimators and other penalized estimators are discussed in [13, 25, 26]. Our work adds to this literature by considering ensembles of models with general weighting and feature structures.

## 2   Preliminaries

In this section, we formally define our weighted estimator and state the main assumption on the weight matrix. Let $f_{\mathrm{nn}} \colon \mathbb{R}^d \to \mathbb{R}^p$ be a pretrained model. Let $\{(\boldsymbol{x}_i, y_i) \colon i = 1, \ldots, n\}$ in $\mathbb{R}^d \times \mathbb{R}$ be the given dataset. Applying $f_{\mathrm{nn}}$ to the raw dataset, we obtain the pretrained features $\boldsymbol{\phi}_i = f_{\mathrm{nn}}(\boldsymbol{x}_i)$ for $i = 1, \ldots, n$ as the resulting neural representations or neural embeddings. In matrix notation, we denote the pretrained feature matrix by $\boldsymbol{\Phi} = [\boldsymbol{\phi}_1, \ldots, \boldsymbol{\phi}_n]^\top \in \mathbb{R}^{n \times p}$. Let $\boldsymbol{W} \in \mathbb{R}^{n \times n}$ be a general weight matrix used for weighting the observations. The weight matrix $\boldsymbol{W}$ is allowed to be asymmetric, in general.

We consider fitting ridge regression on the weighted dataset $(\boldsymbol{W}\boldsymbol{\Phi}, \boldsymbol{W}\boldsymbol{y})$. Given a ridge penalty $\lambda$, the ridge estimator fitted on the weighted dataset is given by:

$$\widehat{\boldsymbol{\beta}}_{\boldsymbol{W},\lambda} := \underset{\boldsymbol{\beta}\in\mathbb{R}^p}{\operatorname{argmin}} \left( \frac{\|\boldsymbol{W}\boldsymbol{y} - \boldsymbol{W}\boldsymbol{\Phi}\boldsymbol{\beta}\|_2^2}{n} + \lambda\|\boldsymbol{\beta}\|_2^2 \right) = (\boldsymbol{\Phi}^\top\boldsymbol{W}^\top\boldsymbol{W}\boldsymbol{\Phi} + n\lambda\boldsymbol{I}_p)^\dagger\boldsymbol{\Phi}^\top\boldsymbol{W}^\top\boldsymbol{W}\boldsymbol{y}. \quad (1)$$

In the definition above, we allow for $\lambda = 0$, in which case the corresponding ridgeless estimator is defined as the limit $\lambda \to 0^+$. For $\lambda < 0$, we use the Moore-Penrose pseudoinverse. An important special case is where $\boldsymbol{W}$ is a diagonal matrix, in which case the above estimator reduces to weighted ridge regression. This type of weight matrix encompasses various applications, such as resampling, bootstrapping, and variance weighting. Our main application in this paper will be subsampling.

For our theoretical results, we assume that the weight matrix $\boldsymbol{W}$ preserves some spectral structure of the feature matrix $\boldsymbol{\Phi}$. This assumption is captured by the condition of *asymptotic freeness* between $\boldsymbol{W}^\top\boldsymbol{W}$ and the feature Gram matrix $\boldsymbol{\Phi}\boldsymbol{\Phi}^\top$. Asymptotic freeness is a concept from free probability theory [64].

**Assumption A** (Weight structure). Let $\boldsymbol{W}^\top\boldsymbol{W}$ and $\boldsymbol{\Phi}\boldsymbol{\Phi}^\top/n$ converge almost surely to bounded operators that are infinitesimally free with respect to $(\overline{\operatorname{tr}}[\cdot], \operatorname{tr}[\boldsymbol{C}(\cdot)])$ for any $\boldsymbol{C}$ independent of $\boldsymbol{W}$ with $\|\boldsymbol{C}\|_{\operatorname{tr}}$ uniformly bounded. Additionally, let $\boldsymbol{W}^\top\boldsymbol{W}$ have a limiting $S$-transform that is analytic on the lower half of the complex plane.

At a high level, Assumption A captures the notion of independence but is adapted for non-commutative random variables of matrices. We provide background on free probability theory and asymptotic freeness in Appendix A.3. Here, we briefly list a series of invertible transformations from free probability to help define the $S$-transform [47]. The Cauchy transform is given by $\mathcal{G}_{\boldsymbol{A}}(z) = \overline{\operatorname{tr}}[(z\boldsymbol{I} - \boldsymbol{A})^{-1}]$. The moment generating series is given by $\mathcal{M}_{\boldsymbol{A}}(z) = z^{-1}\mathcal{G}_{\boldsymbol{A}}(z^{-1}) - 1$. The $S$-transform is given by $\mathcal{S}_{\boldsymbol{A}}(w) = (1 + w^{-1})\mathcal{M}_{\boldsymbol{A}}^{\langle-1\rangle}(w)$. These are the Cauchy transform (negative of the Stieltjes transform), moment generating series, and $S$-transform of $\boldsymbol{A}$, respectively. Here, $\mathcal{M}_{\boldsymbol{A}}^{\langle-1\rangle}$ denotes the inverse under the composition of $\mathcal{M}_{\boldsymbol{A}}$. The notation $\overline{\operatorname{tr}}[\boldsymbol{A}]$ denotes the average trace $\operatorname{tr}[\boldsymbol{A}]/p$ of $\boldsymbol{A} \in \mathbb{R}^{p\times p}$.

The freeness of a pair of matrices $\boldsymbol{A}$ and $\boldsymbol{B}$ means that the eigenvectors of one are completely unaligned or incoherent with those of the other. For example, if $\boldsymbol{A} = \boldsymbol{U}\boldsymbol{R}\boldsymbol{U}^\top$ for a uniformly random unitary matrix $\boldsymbol{U}$ drawn independently of the positive semidefinite $\boldsymbol{B}$ and $\boldsymbol{R}$, then $\boldsymbol{A}$ and $\boldsymbol{B}$ are almost surely asymptotically infinitesimally free [15]. Other well-known examples include Wigner matrices, which are asymptotically free with respect to deterministic matrices [4, Theorem 5.4.5]. Gaussian matrices, where the Gram matrix $\boldsymbol{G} = \boldsymbol{\Phi}\boldsymbol{\Phi}^\top/n = \boldsymbol{U}(\boldsymbol{V}\boldsymbol{V}^\top/n)\boldsymbol{U}^\top$ and any deterministic $\boldsymbol{S}$, are almost surely asymptotically free [47, Chapter 4, Theorem 9]. Although not proven in full generality, it is expected that diagonal matrices are asymptotically free from data Gram matrices constructed using i.i.d. data. In Section 3.2, we will provide additional examples of feature matrices, such as random and kernel features from machine learning, for which our results apply.

Our results involve the notion of degrees of freedom from statistical optimism theory [27, 28]. Degrees of freedom in statistics count the number of dimensions in which a statistical model may vary, which is simply the number of variables for ordinary linear regression. To account for regularization, this notion has been extended to *effective degrees of freedom* (Chapter 3 of [31]). Under some regularity conditions, from Stein's relation [63], the degrees of freedom of a predictor $\widehat{f}$ are measured by the trace of the operators $\boldsymbol{y} \mapsto (\partial/\partial\boldsymbol{y})\widehat{f}(\boldsymbol{\Phi})$. For the ridge estimator $\widehat{\boldsymbol{\beta}}_{\boldsymbol{I},\mu}$ fitted on $(\boldsymbol{\Phi}, \boldsymbol{y})$ with penalty $\mu$, the degrees of freedom is consequently the trace of its prediction operator $\boldsymbol{y} \mapsto \boldsymbol{\Phi}(\boldsymbol{\Phi}^\top\boldsymbol{\Phi} + \mu\boldsymbol{I}_p)^\dagger\boldsymbol{\Phi}^\top\boldsymbol{y}$, which is also referred to as the ridge smoother matrix. That is, $\operatorname{df}(\widehat{\boldsymbol{\beta}}_{\boldsymbol{I},\mu}) = \operatorname{tr}[\boldsymbol{\Phi}^\top\boldsymbol{\Phi}(\boldsymbol{\Phi}^\top\boldsymbol{\Phi} + \mu\boldsymbol{I}_p)^\dagger]$. We denote the normalized degrees of freedom by $\overline{\operatorname{df}} = \operatorname{df}/n$. Note that $\overline{\operatorname{df}}(\widehat{\boldsymbol{\beta}}_{\boldsymbol{I},\mu}) \le \min\{n, p\}/n \le 1$.

Finally, we express our asymptotic results using the asymptotic equivalence relation. Consider sequences $\{\boldsymbol{A}_n\}_{n\ge 1}$ and $\{\boldsymbol{B}_n\}_{n\ge 1}$ of (random or deterministic) matrices (which includes vectors and scalars). We say that $\boldsymbol{A}_n$ and $\boldsymbol{B}_n$ are *equivalent* and write $\boldsymbol{A}_n \simeq \boldsymbol{B}_n$ if $\lim_{p\to\infty}|\operatorname{tr}[\boldsymbol{C}_n(\boldsymbol{A}_n - \boldsymbol{B}_n)]| = 0$ almost surely for any sequence $\boldsymbol{C}_n$ of matrices with bounded trace norm such that $\limsup\|\boldsymbol{C}_n\|_{\operatorname{tr}} < \infty$ as $n \to \infty$. Our forthcoming results apply to a sequence of problems indexed by $n$. For notational simplicity, we omit the explicit dependence on $n$ in our statements.

# 3 Implicit regularization paths

We begin by characterizing the implicit regularization induced by weighted pretrained features. We will show that the degrees of freedom of the unweighted estimator $\widehat{\boldsymbol{\beta}}_{\boldsymbol{I},\mu}$ on the full data $(\boldsymbol{\Phi}, \boldsymbol{y})$ with regularization parameter $\mu$ are equal to the degrees of freedom of the weighted estimator $\widehat{\boldsymbol{\beta}}_{\boldsymbol{W},\lambda}$ for some regularization parameter $\lambda$. For estimator equivalence, our data-dependent set of weighted ridge estimators $(\boldsymbol{W}, \lambda)$ that connect to the unweighted ridge estimator $(\boldsymbol{I}, \mu)$ is defined in terms of "matching" effective degrees of freedom of component estimators in the set.

To state the upcoming result, denote the Gram matrix of the weighted data as $\boldsymbol{G}_{\boldsymbol{W}} = \boldsymbol{W}\boldsymbol{\Phi}\boldsymbol{\Phi}^{\top}\boldsymbol{W}^{\top}/n$ and the Gram matrix of the unweighted data as $\boldsymbol{G}_{\boldsymbol{I}} = \boldsymbol{\Phi}\boldsymbol{\Phi}^{\top}/n$. Furthermore, let $\lambda_{\min}^{+}(\boldsymbol{A})$ denote the minimum positive eigenvalue of a symmetric matrix $\boldsymbol{A}$.

**Theorem 1** (Implicit regularization of weighted representations). *For $\boldsymbol{G}_{\boldsymbol{I}} \in \mathbb{R}^{n \times n}$, suppose that the weight matrix $\boldsymbol{W} \in \mathbb{R}^{n \times n}$ satisfies Assumption A and $\limsup \|\boldsymbol{y}\|_{2}^{2}/n < \infty$ as $n \to \infty$. For any $\mu > -\liminf_{n\to\infty} \lambda_{\min}^{+}(\boldsymbol{G}_{\boldsymbol{I}})$, let $\lambda > -\lambda_{\min}^{+}(\boldsymbol{G}_{\boldsymbol{W}})$ be given by the following equation:*

$$\lambda = \mu/\mathcal{S}_{\boldsymbol{W}^{\top}\boldsymbol{W}}(-\overline{\mathsf{df}}(\widehat{\boldsymbol{\beta}}_{\boldsymbol{I},\mu})), \tag{2}$$

*where $\mathcal{S}_{\boldsymbol{W}^{\top}\boldsymbol{W}}$ is the S-transform of the operator $\boldsymbol{W}^{\top}\boldsymbol{W}$. Then, as $n \to \infty$, it holds that:*

$$\mathsf{df}(\widehat{\boldsymbol{\beta}}_{\boldsymbol{W},\lambda}) \simeq \mathsf{df}(\widehat{\boldsymbol{\beta}}_{\boldsymbol{I},\mu}) \quad and \quad \widehat{\boldsymbol{\beta}}_{\boldsymbol{W},\lambda} \simeq \widehat{\boldsymbol{\beta}}_{\boldsymbol{I},\mu}. \tag{3}$$

In other words, to achieve a target regularization of $\mu$ on the unweighted data, Theorem 1 provides a method to compute the regularization penalty $\lambda$ with given weights $\boldsymbol{W}$ from the available data using (2). The weighted estimator then has asymptotically the same degrees of freedom as the unweighted estimator. This means that the level of effective regularization of the two estimators is the same. Moreover, the estimators themselves are structurally equivalent; that is, $\boldsymbol{c}^{\top}(\widehat{\boldsymbol{\beta}}_{\boldsymbol{W},\lambda} - \widehat{\boldsymbol{\beta}}_{\boldsymbol{I},\mu}) \xrightarrow{\text{a.s.}} 0$ for every constant vector $\boldsymbol{c}$ with bounded norm. The estimator equivalence in Theorem 1 is a "first-order" result, while we will also characterize the "second-order" effects in Section 4.

The notable aspect of Theorem 1 is its generality. The equivalence results hold for a wide range of weight matrices and allow for negative values for the regularization levels. Furthermore, we have not made any direct assumptions about the feature matrix $\boldsymbol{\Phi}$, the weight matrix $\boldsymbol{W}$, and the response vector $\boldsymbol{y}$ (other than mild bounded norms). The main underlying ingredient is the asymptotic freeness between $\boldsymbol{W}$ and $\boldsymbol{\Phi}$, which we then exploit using tools developed in [39] in the context of feature sketching. We discuss special cases of interest for $\boldsymbol{W}$ and $\boldsymbol{\Phi}$ in the upcoming Sections 3.1 and 3.2.

## 3.1 Examples of weight matrices

There are two classes of weighting matrices that are of practical interest:

- **Non-diagonal weighting matrices.** One can consider observation sketching, which involves some random linear combinations of the rows of the data matrix. Such observation sketching is beneficial for privacy, as it scrambles the rows of the data matrix, which may contain identifiable information about individuals. It also helps in reducing the effect of non-i.i.d. data that arise in time series or spatial data, where one wants to smooth away the impact of irregularities or non-stationarity.

- **Diagonal weighting matrices.** When observations are individually weighted, $\boldsymbol{W}$ is a diagonal matrix, which includes scenarios such as resampling, bootstrapping, and subsampling. Note that even with subsampling, one can have a non-binary diagonal weighting matrix. For example, one can consider sampling with replacement or sampling with a particular distribution, which yields non-binary diagonal weighting matrices. Other examples of non-binary diagonal weighting matrices include inverse-variance weighting sampling to mitigate the effects of heterogeneous variations if the responses have different variances for different units.

In general, the set of equivalent weighted estimators depends on the corresponding $S$-transform as in (2), and it can be numerically evaluated. When focusing on subsampling without replacement, the data-dependent path for equivalent estimators with associated subsampling and regularization levels can be explicitly characterized in the following result by analyzing the $S$-transform of subsampling operators.

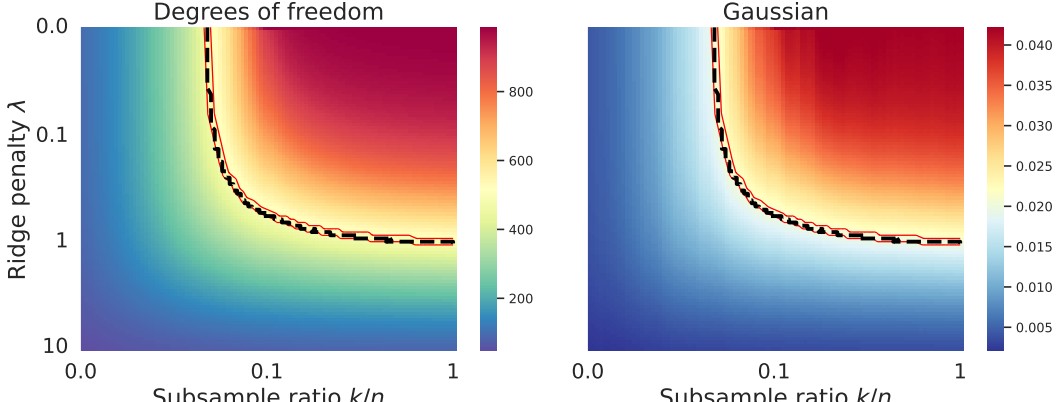

Figure 1: Equivalence under subsampling. The left panel shows the heatmap of degrees of freedom, and the right panel shows the random projection $\mathbb{E}_{\boldsymbol{W}}[\boldsymbol{a}^\top \widehat{\boldsymbol{\beta}}_{\boldsymbol{W},\lambda}]$ where $\boldsymbol{a} \sim \mathcal{N}(\boldsymbol{0}_p, \boldsymbol{I}_p/p)$. In both heatmaps, the red color lines indicate the predicted paths using Equation (4), and the black dashed lines indicate the empirical paths by matching empirical degrees of freedom. The data is generated according to Appendix F.1 with $n = 10000$ and $p = 1000$, and the results are averaged over $M = 100$ random weight matrices $\boldsymbol{W}$.

**Theorem 2** (Regularization paths due to subsampling). *For a subsampling matrix $\boldsymbol{W}^{(k)}$ consisting of $k$ unit diagonal entries, the path (2) in terms of $(k, \lambda)$ simplifies to:*

$$(1 - \mathsf{df}/n) \cdot (1 - \lambda/\mu) = (1 - k/n), \tag{4}$$

*where we denote by $\mathsf{df} = \mathsf{df}(\widehat{\boldsymbol{\beta}}_{\boldsymbol{I},\mu}) = \mathsf{df}(\widehat{\boldsymbol{\beta}}_{\boldsymbol{W},\lambda})$ for notational simplicity.*

The relation (4) is remarkably simple, yet quite general! It provides an interplay between the normalized target complexity $\mathsf{df}/n$, regularization inflation $\lambda/\mu$, and subsample fraction $k/n$:

$$(1 - \text{normalized complexity}) \cdot (1 - \text{regularization inflation}) = (1 - \text{subsample fraction}). \tag{5}$$

Since the normalized target complexity and subsample fraction are no greater than one, (5) also implies that the regularization level $\lambda$ for the subsample estimator is always lower than the regularization level $\mu$ for the full estimator. In other words, subsampling induces (positive) implicit regularization, reducing the need for explicit ridge regularization. This is verified numerically in Figure 1.

For a fixed target regularization amount $\mu$, the degrees of freedom $\mathsf{df}(\widehat{\boldsymbol{\beta}}_{\boldsymbol{I},\mu})$ of the ridge estimator on full data is fixed. Thus, we can observe that the path in the $(k/n, \lambda)$-plane is a line. There are two extreme cases: (1) when the subsample size $k$ is close to $n$, we have $\mu \approx \lambda$; and (2) when the subsample size is near 0, we have $\mu \approx \infty$. When $\lambda = 0$, the effective regularization level $\lambda$ is such that $\mathsf{df}(\widehat{\boldsymbol{\beta}}_{\boldsymbol{W}^{(k)},\lambda}) = \mathsf{df}(\widehat{\boldsymbol{\beta}}_{\boldsymbol{I},\mu}) = k$, which we find to be a neat relation!

Beyond subsampling without replacement, one can also consider other subsample matrixs. For example, for bootstrapping $k$ entries, we observe a similar equivalent path in Figure 5. Additionally, for random sample reweighting, as shown in Figure 6, we also observe certain equivalence behaviors of degrees of freedom. This indicates that Theorem 1 also applies to more general weighting schemes.

### 3.2 Examples of feature matrices

As mentioned in Section 2, when the feature matrix $\boldsymbol{\Phi}$ consists of i.i.d. Gaussian features, any deterministic matrix $\boldsymbol{W}$ satisfies the condition stated in Assumption A. However, our results are not limited to Gaussian features. In this section, we will consider more general families of features commonly analyzed in machine learning and demonstrate the applicability of our results to them.

*(1) Linear features.* As a first example, we consider linear features composed of (multiplicatively) transformed i.i.d. entries with sufficiently bounded moments by a deterministic covariance matrix.

**Proposition 3** (Regularization paths with linear features). Suppose the feature $\phi$ can be decomposed as $\phi = \boldsymbol{\Sigma}^{1/2}\boldsymbol{z}$, where $\boldsymbol{z} \in \mathbb{R}^p$ contains i.i.d. entries $z_i$ for $i = 1, \ldots, p$ with mean 0, variance 1, and satisfies $\mathbb{E}[|z_i|^{4+\mu}] \leq M_\mu < \infty$ for some $\mu > 0$ and a constant $M_\mu$, and $\boldsymbol{\Sigma} \in \mathbb{R}^{p \times p}$ is a deterministic

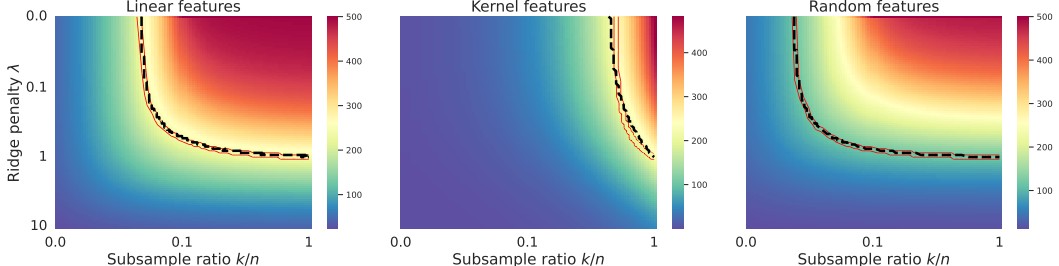

Figure 2: Equivalence of degrees of freedom for various feature structures under subsampling. The three panels correspond to linear features, random features with ReLU activation function (2-layer), and kernel features (polynomial kernel with degree 3 and without intercept), respectively. In all heatmaps, the red color lines indicate the predicted paths using Equation (4), and the black dashed lines indicate the empirical paths by matching the empirical degrees of freedom. The data is generated according to Appendix F.1 with $n = 5000$ and $p = 500$, and the results are averaged over $M = 100$ random weight matrices $\boldsymbol{W}$.

symmetric matrix with eigenvalues uniformly bounded between constants $r_{\min} > 0$ and $r_{\max} < \infty$. Then, as $n, p \to \infty$ such that $p/n \to \gamma > 0$, the equivalences in (3) hold along the path (4).

Features of this type are common in random matrix theory [8] and in a wide range of applications, including statistical physics [14, 60], high-dimensional statistics [22, 55, 58], machine learning [18], among others. The generalized path (2) in Theorem 2 recovers the path in Proposition 4 of [50]. Although the technique in this paper is quite different and more general than that of [50].

*(2) Kernel features.* As the second example, Theorem 2 also applies to kernel features. Kernel features are a generalization of linear features and lift the input feature space to a high- or infinite-dimensional feature space by applying a feature map $\boldsymbol{x} \mapsto \boldsymbol{\phi}(\boldsymbol{x})$. Kernel methods use the kernel function $K(\boldsymbol{x}_i, \boldsymbol{x}_j) = \langle \boldsymbol{\phi}(\boldsymbol{x}_i), \boldsymbol{\phi}(\boldsymbol{x}_j) \rangle$ to compute the inner product in the lifted space.

**Proposition 4** (Regularization paths with kernel features). Suppose the same conditions as in Proposition 3 and the kernel function is of the form $K(\boldsymbol{x}_i, \boldsymbol{x}_j) = g(\|\boldsymbol{x}_i\|_2^2/p, \langle \boldsymbol{x}_i, \boldsymbol{x}_j \rangle/p, \|\boldsymbol{x}_j\|_2^2/p)$, where $g$ is $\mathcal{C}^1$ around $(\tau, \tau, \tau)$ and $\mathcal{C}^3$ around $(\tau, 0, \tau)$ and $\tau := \lim_{p \to \infty} \text{tr}[\boldsymbol{\Sigma}]/d$. Then, as $n \to \infty$, the equivalences in (3) hold in probability along the path (4).

The assumption in Proposition 4 is commonly used in the risk analysis of kernel ridge regression [9, 19, 29, 57], among others. Here, $\mathcal{C}^k$ denotes the class of functions that are $k$-times continuously differentiable. It includes neural tangent kernels (NTKs) as a special case. Proposition 4 confirms Conjecture 8 of [50] for these types of kernel functions.

*(3) Random features.* Finally, we consider random features that were introduced by [56] as a way to scale kernel methods to large datasets. Linked closely to two-layer neural networks [46], the random feature model has $f_{\text{nn}}(\boldsymbol{x}) = \sigma(\boldsymbol{F}\boldsymbol{x})$, where $\boldsymbol{F} \in \mathbb{R}^{d \times p}$ is some randomly initialized weight matrix, and $\sigma : \mathbb{R} \to \mathbb{R}$ is a nonlinear activation function applied element-wise to $\boldsymbol{F}\boldsymbol{x}$.

**Proposition 5** (Regularization paths with random features). Suppose $\boldsymbol{x}_i \sim \mathcal{N}(\boldsymbol{0}, \boldsymbol{\Sigma})$ and the activation function $\sigma : \mathbb{R} \to \mathbb{R}$ is differentiable almost everywhere and there are constants $c_0$ and $c_1$ such that $|\sigma(x)|, |\sigma'(x)| \le c_0 e^{c_1 x}$, whenever $\sigma'(x)$ exists. Then, as $n, p, d \to \infty$ such that $p/n \to \gamma > 0$ and $d/n \to \xi > 0$, the equivalences in (3) hold in probability along the path (4).

As mentioned in the related work, random feature models have recently been used as a standard model to study various generalization phenomena observed in neural networks theoretically [1, 46]. Proposition 5 resolves Conjecture 7 of [50] under mild regularity conditions on the activation function.

It is worth noting that the prior works mentioned above, including [50], have focused on first characterizing the risk asymptotics in terms of various population quantities for each of the cases above. In contrast, our work in this paper deviates from these approaches by not expressing the risk in population quantities but rather by directly relating the estimators at different regularization levels. In the next section, we will explore the relationship between their squared prediction risks.

# 4 Prediction risk asymptotics and risk estimation

The results in the previous section provide first-order equivalences of the estimators, which are related to the bias of the estimators. In practice, we are also interested in the predictive performance of the estimators. In this section, we investigate the second-order equivalence of weighting and ridge regularization through ensembling. Specifically, we show that aggregating estimators fitted on different weighted datasets also reduces the additional variance. Furthermore, the prediction risks of the full-ensemble weighted estimator and the unweighted estimator also match along the path.

Before presenting our risk equivalence result, we first introduce some additional notation. Assume there are $M$ i.i.d. weight matrices $\boldsymbol{W}_1, \ldots, \boldsymbol{W}_M \in \mathbb{R}^{n \times n}$. The $M$-*ensemble* estimator is defined as:

$$\widehat{\boldsymbol{\beta}}_{\boldsymbol{W}_{1:M}, \lambda} = M^{-1} \sum_{m=1}^{M} \widehat{\boldsymbol{\beta}}_{\boldsymbol{W}_m, \lambda}, \tag{6}$$

and its performance is quantified by the conditional squared prediction risk, given by:

$$R(\widehat{\boldsymbol{\beta}}_{\boldsymbol{W}_{1:M}, \lambda}) = \mathbb{E}_{\boldsymbol{x}_0, y_0}[(y_0 - \boldsymbol{\phi}_0^\top \widehat{\boldsymbol{\beta}}_{\boldsymbol{W}_{1:M}, \lambda})^2 \mid \boldsymbol{\Phi}, \boldsymbol{y}, \{\boldsymbol{W}_m\}_{m=1}^M], \tag{7}$$

where $(\boldsymbol{x}_0, y_0)$ is a test point sampled independently from some distribution $P_{\boldsymbol{x}_0, y_0}$ that may be different from the training distribution $P_{\boldsymbol{x}, y}$, and $\boldsymbol{\phi}_0 = f_{\mathrm{nn}}(\boldsymbol{x}_0)$ is the pretrained feature at the test point. The covariance matrix of the test features $\boldsymbol{\phi}_0$ is denoted by $\boldsymbol{\Sigma}_0$. When $P_{\boldsymbol{x}_0, y_0} = P_{\boldsymbol{x}, y}$, we refer to it as the in-distribution risk. On the other hand, when $P_{\boldsymbol{x}_0, y_0}$ differs from $P_{\boldsymbol{x}, y}$, we refer to it as the out-of-distribution risk. Note that the conditional risk $R_M$ is a scalar random variable that depends on both the dataset $(\boldsymbol{\Phi}, \boldsymbol{y})$ and the weight matrix $\boldsymbol{W}_m$ for $m \in [M]$. Our goal in this section is to analyze the prediction risk of the ensemble estimator (6) for any ensemble size $M$.

**Theorem 6** (Risk equivalence along the path). *Under the setting of Theorem 1, assume that the operator norm of $\boldsymbol{\Sigma}_0$ is uniformly bounded in $p$ and that each response variable $y_i$ for $i = 1, \ldots, n$ has mean $0$ and satisfies $\mathbb{E}[|y_i|^{4+\mu}] \leq M_\mu < \infty$ for some $\mu, M_\mu > 0$. Then, along the path (4),*

$$R(\widehat{\boldsymbol{\beta}}_{\boldsymbol{W}_{1:M}, \lambda}) \simeq R(\widehat{\boldsymbol{\beta}}_{\boldsymbol{I}, \mu}) + \frac{C}{M} \overline{\mathrm{tr}}[(\boldsymbol{G}_{\boldsymbol{I}} + \mu \boldsymbol{I})^\dagger \boldsymbol{y} \boldsymbol{y}^\top (\boldsymbol{G}_{\boldsymbol{I}} + \mu \boldsymbol{I}_n)^\dagger], \tag{8}$$

*where the constant $C$ is given by:*

$$C = -\partial \mu / \partial \lambda \cdot \lambda^2 \mathcal{S}'_{\boldsymbol{W}^\top \boldsymbol{W}}(-\mathsf{df}(\widehat{\boldsymbol{\beta}}_{\boldsymbol{I}, \mu})) \overline{\mathrm{tr}}[(\boldsymbol{G}_{\boldsymbol{I}} + \mu \boldsymbol{I})^\dagger (\boldsymbol{\Phi} \boldsymbol{\Sigma}_0 \boldsymbol{\Phi}^\top / n)(\boldsymbol{G}_{\boldsymbol{I}} + \mu \boldsymbol{I})^\dagger]. \tag{9}$$

At a high level, Theorem 6 provides a bias-variance-like risk decomposition for both the squared risks of weighted ensembles. The risk of the weighted predictor is equal to the risk of the unweighted equivalent implicit ridge regressor (bias) plus a term due to the randomness due to weighting (variance). The inflation factor $C$ controls the magnitude of this term, and it decreases at a rate of $1/M$ as the ensemble size $M$ increases (see Figure 7 for a numerical verification of this rate). Therefore, by using a resample ensemble with a sufficiently large size $M$, we can retain the statistical properties of the full ridge regression while reducing memory usage and increasing parallelization.

Theorem 6 extends the risk equivalence results in [50, 52]. Compared to previous results, Theorem 6 provides a broader risk equivalence that holds for general weight and feature matrices, as well as an arbitrary ensemble size $M$. It is important to note that Theorem 6 holds even when the test distribution differs from the training data, making it applicable to out-of-distribution risks. Furthermore, our results do not rely on any specific distributional assumptions for the response vector, making them applicable in a model-free setting. The key idea behind this result is to exploit asymptotic freeness between the subsample and data matrices. Next, we will address the question of optimal tuning.

## 4.1 Optimal oracle tuning

As in Theorem 2, we next analyze various properties related to optimal subsampling weights and their implications for the risk of optimal ridge regression. Recall that the subsampling matrix $\boldsymbol{W}^{(k)}$ is a diagonal matrix with $k \in \{1, \ldots, n\}$ nonzero diagonal entries, which is parameterized by the subsample size $k$. Note that the optimal regularization parameter $\mu^*$ for the full data ($\boldsymbol{W}^{(k)} = \boldsymbol{I}$ or $k = n$) is a function of the distribution of pretrained data and the test point. Based on the risk equivalence in Theorem 6, there exists an optimal path of $(k, \lambda)$ with the corresponding full-ensemble

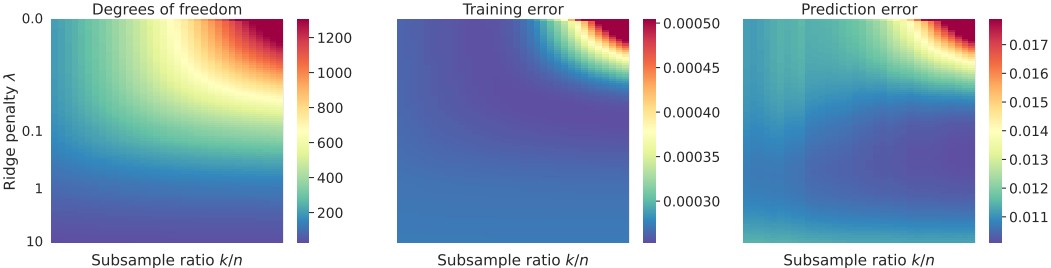
Figure 3: Equivalence in pretrained features of pretrained ResNet-50 on Flowers-102 datasets.

estimator $\widehat{\boldsymbol{\beta}}_{\boldsymbol{W}_{1:\infty}^{(k)},\lambda} := \lim_{M\to\infty} \widehat{\boldsymbol{\beta}}_{\boldsymbol{W}_{1:M}^{(k)},\lambda}$ that achieves the optimal predictive performance at $(n,\mu^*)$. In particular, the ridgeless ensemble with $\lambda^* = 0$ happens to be on the path. From previous work [25, 50], the optimal subsample size $k^*$ for $\lambda^* = 0$ has the property that $k^* \leq p$ under linear features. We show in the following that this property can be extended to include general features.

**Proposition 7** (Optimal subsample ratio). Assume the subsampling matrix $\boldsymbol{W}$ as defined in Theorem 2. Let $\mu^* = \operatorname{argmin}_{\mu\geq 0} R(\widehat{\boldsymbol{\beta}}_{\boldsymbol{W}_{1:\infty}^{(k)},\mu})$. Then the corresponding subsample size satisfies:

$$k^* = \mathsf{df}(\widehat{\boldsymbol{\beta}}_{\boldsymbol{W}_{1:\infty}^{(k)},\mu^*}) \leq \operatorname{rank}(\boldsymbol{G_I}). \tag{10}$$

The optimal subsample size $k^*$ obtained from Proposition 7 is asymptotically optimal. For linear features, in the underparameterized regime where $n > p$, [25, 50] show that the optimal subsample size $k^*$ is asymptotically no larger than $p$. This result is covered by Proposition 7 by noting that $\operatorname{rank}(\boldsymbol{G_I}) \leq p$ under linear features. It is interesting and somewhat surprising to note that in the underparameterized regime (when $p \leq n$), we do not need more than $p$ observations to achieve the optimal risk. In this sense, the optimal subsampled dataset is always overparameterized.

When the limiting risk profiles $\mathscr{R}(\gamma,\psi,\mu) := \lim_{p/n\to\gamma,p/k\to\psi} R(\widehat{\boldsymbol{\beta}}_{\boldsymbol{W}_{1:\infty}^{(k)},\mu})$ exist for subsample ensembles, the limiting risk of the optimal ridge predictor $\inf_{\mu\geq 0} \mathscr{R}(\gamma,\gamma,\mu)$ is monotonically decreasing in the limiting sample aspect ratio $\gamma$ [50]. This also (provably) confirms the sample-wise monotonicity of optimally-tuned risk for general features in an asymptotic sense [48]. Due to the risk equivalence in Theorem 6, for any $\mu > 0$, there exists $\psi$ such that $\mathscr{R}(\gamma,\gamma,\mu) = \mathscr{R}(\gamma,\psi,0)$. This implies that $\inf_{\mu\geq 0} \mathscr{R}(\gamma,\gamma,\mu) = \inf_{\psi\geq\gamma} \mathscr{R}(\gamma,\psi,0)$. In other words, tuning over subsample sizes with sufficiently large ensembles is equivalent to tuning over the ridge penalty on the full data.

### 4.2 Data-dependent tuning

As suggested by Proposition 7, the optimal subsample size is smaller than the rank of the Gram matrix. This result has important implications for real-world datasets where the number of observations ($n$) is much larger than the number of features ($p$). In such cases, instead of using the entire dataset, we can efficiently build small ensembles with a subsample size $k \leq p$. This approach is particularly beneficial when $n$ is significantly higher than $p$, for example, when $n = 1000p$. By fitting ensembles with only $M = 100$ base predictors, we can potentially reduce the computational burden while still achieving optimal predictive performance. Furthermore, this technique can be especially valuable in scenarios where computational resources are limited or when dealing with massive datasets that cannot be easily processed in their entirety.

In the following, we propose a method to determine the optimal values of the regularization parameter $\mu^*$ for the full ridge regression, as well as the corresponding subsample size $k^*$ and the optimal ensemble size $M^*$. According to Theorem 6, the optimal value of $M^*$ is theoretically infinite. However, in practice, the prediction risk of the $M$-ensemble predictor decreases at a rate of $1/M$ as $M$ increases. Therefore, it is important to select a suitable value of $M$ that achieves the desired level of performance while considering computational constraints and the specified error budget. By carefully choosing an appropriate $M$, we can strike a balance between model accuracy and efficiency, ensuring that the subsampled neural representations are effectively used in downstream tasks.

Consider a grid of subsample size $\mathcal{K}_n \subseteq \{1,\dots,n\}$; for instance, $\mathcal{K}_n = \{0,k_0,2k_0,\dots,n\}$ where $k_0$ is a subsample size unit. For a prespecified subsample size $k \in \mathcal{K}_n$ and ensemble size $M_0 \in$

---

**Algorithm 1** Meta-algorithm for tuning of ensemble sizes and subsample matrices.

---

**Input:** A dataset $\mathcal{D}_n = \{(\boldsymbol{x}_i, y_i) \in \mathbb{R}^p \times \mathbb{R} : 1 \leq i \leq n\}$, a regularization parameter $\lambda$, a class of subsample matrix distribution $\mathcal{P}_n = \{P_k\}_{k \in \mathcal{K}_n}$, a ensemble size $M_0 \geq 2$ for risk estimation, and optimality tolerance parameter $\delta$.

1: Build ensembles $\widehat{\boldsymbol{\beta}}_{\boldsymbol{W}_{1:M_0}^{(k)}, \lambda}$ with $M_0$ base estimators, where $\boldsymbol{W}_1^{(k)}, \ldots, \boldsymbol{W}_{M_0}^{(k)} \overset{\text{i.i.d.}}{\sim} P_k$ for each $k \in \mathcal{K}_n$.

2: Estimate the prediction risk of $\widehat{\boldsymbol{\beta}}_{\boldsymbol{W}_{1:M_0}^{(k)}, \lambda}$ with $\widehat{R}_{m,k}$ by CV methods such as CGCV [13], for $k \in \mathcal{K}_n$ and $m = 1, \ldots, M_0$.

3: Extrapolate the risk estimations $\widehat{R}_{m,k}$ for $m > M_0$ using (11) and (12).

4: Select a subsample size $\widehat{k} \in \text{argmin}_{k \in \mathcal{K}_n} \widehat{R}_{\infty, k}$. that minimizes the extrapolated estimates.

5: Select an ensemble size $\widehat{M} \in \text{argmin}_{m \in \mathbb{N}} \mathbb{1}\{\widehat{R}_{m,\widehat{k}} > \widehat{R}_{\infty, \widehat{k}} + \delta\}$ for the $\delta$-optimal risk.

6: If $\widehat{M} > M_0$, fit a $\widehat{M}$-ensemble estimator $\widehat{\boldsymbol{\beta}}_{\boldsymbol{W}_{1:\widehat{M}}^{(\widehat{k})}, \lambda}$.

**Output:** Return the tuned estimator $\widehat{\boldsymbol{\beta}}_{\boldsymbol{W}_{1:\widehat{M}}^{(\widehat{k})}, \lambda}$, and the risk estimators $\widehat{R}_{M,k}$ for all $M, k$.

---

$\mathbb{N}$, suppose we have multiple risk estimates $\widehat{R}_m$ of $R_m$ for $m = 1, \ldots, M_0$. The squared risk decomposition [51, Eq (7)] along with the equivalence path (8) implies that $R_m = m^{-1} R_1 + \left(1 - m^{-1}\right) R_\infty$, for $m = 1, \ldots, M_0$. Summing these equations yields $\sum_{m=1}^{M_0} R_m = \sum_{m=1}^{M_0} \frac{1}{m} R_1 + \sum_{m=1}^{M_0} \left(1 - m^{-1}\right) R_\infty$. Thus, we can estimate $R_\infty$ by:

$$\widehat{R}_\infty = \left( \sum_{m=1}^{M_0} \widehat{R}_m - \sum_{m=1}^{M_0} m^{-1} \widehat{R}_1 \right) / \sum_{m=1}^{M_0} \left(1 - m^{-1}\right). \tag{11}$$

Then, the extrapolated risk estimates $\widehat{R}_m$ (with $m > M_0$) are defined as:

$$\widehat{R}_m := m^{-1} \widehat{R}_1 + \left(1 - m^{-1}\right) \widehat{R}_\infty \quad \text{for} \quad m > M_0. \tag{12}$$

The meta-algorithm that implements the above cross-validation procedure is provided in Algorithm 1. To efficiently tune the parameters of ridge ensembles, we use and combine the corrected generalized cross-validation (CGCV) method [13] and the extrapolated cross-validation (ECV) method [26]. The improved CV method is implemented in the Python library [24].

### 4.3 Validation on real-world datasets

In this section, we present numerical experiments to validate our theoretical results on real-world datasets. Figure 3 provides evidence supporting Assumption A on pretrained features extracted from commonly used neural networks applied to real-world datasets. The first panel of the figure demonstrates the equivalence of degrees of freedom for these pretrained features. Furthermore, we also observe consistent behavior across different neural network architectures and different datasets (see Figures 8 and 9). Remarkably, the path of equivalence can be accurately predicted, offering valuable insight into the underlying dynamics of these models. This observation suggests that the pretrained features from widely used neural networks exhibit similar properties when applied to real-world data, regardless of the specific architecture employed. The ability to predict the equivalence path opens up new possibilities for optimizing the performance of these models in practical applications.

One implication of the equivalence results explored in Theorems 1 and 6 is that instead of tuning for the full ridge penalty $\mu$ on the large datasets, we can fix a small value of the ridge penalty $\lambda$, fit subsample ridge ensembles, and tune for an optimal subsample size $k$. To illustrate the validity of the tuning procedure described in Algorithm 1, we present both the actual prediction errors and their estimates by Algorithm 1 in Figure 4. We observe that the risk estimates closely match the prediction risks at different ensemble sizes across different datasets. Even with a subsampling ratio $k/n$ of 0.01 and a sufficiently large $M$, the risk estimate is close to the optimal risk. A smaller subsample size could also yield even smaller prediction risk in certain datasets.

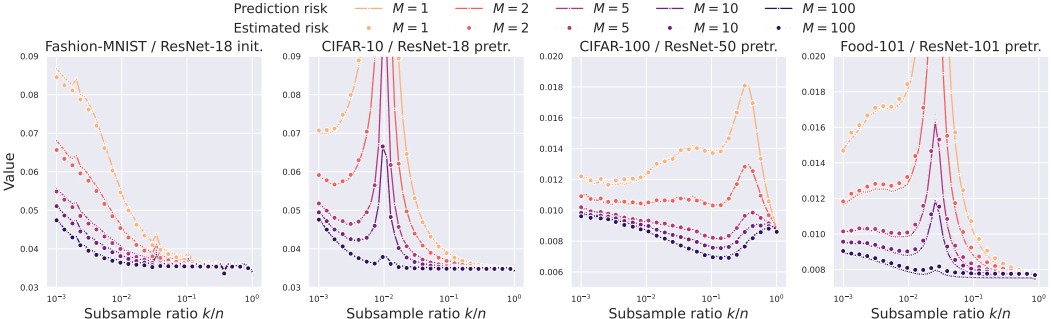

Figure 4: Risk estimation by corrected and extrapolated generalized cross-validation. The risk estimates are computed based on $M_0 = 25$ base estimators using Algorithm 1 with $\lambda = 10^{-3}$.

## 5 Limitations and outlook

While our results are quite general in terms of applying to a wide variety of pretrained features, they are limited in that they only apply to ridge regression fitted on the pretrained features. The key challenge for extending the analysis based on Assumption A to general estimators beyond ridge regression is the characterization of the effect of subsampling general resolvents as additional ridge regularization. To extend to generalized linear models, one approach is to view the optimization as iteratively reweighted least squares [38] in combination with the current results. Another approach is to combine our results with the techniques in [41] to obtain deterministic equivalents for the Hessian, enabling an understanding of implicit regularization due to subsampling beyond linear models.

Beyond implicit regularization due to subsampling, there are other forms of implicit regularization, such as algorithmic regularization due to early stopping in gradient descent [2, 3, 49], dropout regularization [62, 65], among others. In some applications, multiple forms of implicit regularization are present simultaneously. For instance, during a mini-batch gradient step, implicit regularization arises from both iterative methods and mini-batch subsampling. The results presented in this paper may help to make explicit the combined effect of various forms of implicit regularization.

## Acknowledgments and Disclosure of Funding

We thank Benson Au, Daniel LeJeune, Ryan Tibshirani, and Alex Wei for the helpful conversations surrounding this work. We also thank the anonymous reviewers for their valuable feedback and suggestions.

We acknowledge the computing support the ACCESS allocation MTH230020 provided for some of the experiments performed on the Bridges2 system at the Pittsburgh Supercomputing Center. The code for reproducing the results of this paper can be found at https://jaydu1.github.io/overparameterized-ensembling/weighted-neural.

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

# Appendix

This serves as an appendix to the paper "Implicit Regularization Paths of Weighted Neural Representations." The beginning (unlabeled) section of the appendix provides an organization for the appendix, followed by a summary of the general notation used in both the paper and the appendix. Any other specific notation is explained inline where it is first used.

**Organization**

- In Appendix A, we provide a brief technical background on free probability theory and various transforms that we need and collect known asymptotic ridge equivalents that we use in our proofs.
- In Appendix B, we present proofs of the theoretical results in Section 3 (Theorems 1 and 2 and Propositions 3–5).
- In Appendix C, we present proofs of the theoretical results in Section 4 (Theorem 6 and Proposition 7).
- In Appendix D, we provide additional illustrations for the results in Section 3 (Figures 5 and 6).
- In Appendix E, we provide additional illustrations for the results in Section 4 (Figures 7–9), including our meta-algorithm for tuning (Algorithm 1) that is not included in the main text due to space constraints.
- In Appendix F, we provide additional details on the experiments in both Section 3 and Section 4.

**Notation**

We use blackboard letters to denote some special sets: $\mathbb{N}$ denotes the set of natural numbers, $\mathbb{R}$ denotes the set of real numbers, $\mathbb{R}_+$ denotes the set of positive real numbers, $\mathbb{C}$ denotes the set of complex numbers, $\mathbb{C}^+$ denotes the set of complex numbers with positive imaginary part, and $\mathbb{C}^-$ denotes the set of complex numbers with negative imaginary part. We use $[n]$ to denote the index set $\{1, 2, \ldots, n\}$.

We denote scalars and vectors using lower-case letters and matrices using upper-case letters. For a vector $\boldsymbol{\beta}$, $\boldsymbol{\beta}^\top$ denotes its transpose, and $\|\boldsymbol{\beta}\|_2$ denotes its $\ell_2$ norm. For a pair of vectors $\boldsymbol{u}$ and $\boldsymbol{v}$, $\langle \boldsymbol{u}, \boldsymbol{v} \rangle$ denotes their inner product. For a matrix $\boldsymbol{X} \in \mathbb{R}^{n \times p}$, $\boldsymbol{X}^\top \in \mathbb{R}^{p \times n}$ denotes its transpose, and $\boldsymbol{X}^\dagger \in \mathbb{R}^{p \times n}$ denotes its Moore-Penrose inverse. For a square matrix $\boldsymbol{A} \in \mathbb{R}^{p \times p}$, $\mathrm{tr}[\boldsymbol{A}]$ denotes its trace, $\overline{\mathrm{tr}}[\boldsymbol{A}]$ denotes its average trace $\mathrm{tr}[\boldsymbol{A}]/p$, and $\boldsymbol{A}^{-1}$ denotes its inverse, provided that $\boldsymbol{A}$ is invertible. For a symmetric matrix $\boldsymbol{A}$, $\lambda_{\min}^+(\boldsymbol{A})$ denotes its minimum nonzero eigenvalue. For a positive semidefinite matrix $\boldsymbol{G}$, $\boldsymbol{G}^{1/2}$ denotes its principal square root. For a matrix $\boldsymbol{X}$, we denote by $\|\boldsymbol{X}\|_{\mathrm{op}}$ its operator norm with respect to the $\ell_2$ vector norm. It is also the spectral norm of $\boldsymbol{X}$. For a matrix $\boldsymbol{X}$, we denote by $\|\boldsymbol{X}\|_{\mathrm{tr}}$ its trace norm. It is given by $\mathrm{tr}[(\boldsymbol{X}^\top \boldsymbol{X})^{1/2}]$, and is also the nuclear norm $\boldsymbol{X}$. We denote $p \times p$ identity matrix by $\boldsymbol{I}_p$, or simply by $\boldsymbol{I}$ when it is clear from the context.

For symmetric matrices $\boldsymbol{A}$ and $\boldsymbol{B}$, we use $\boldsymbol{A} \preceq \boldsymbol{B}$ to denote the Loewner ordering to mean that $\boldsymbol{A} - \boldsymbol{B}$ is a positive semidefinite matrix. For two sequences of matrices $\boldsymbol{A}_p$ and $\boldsymbol{B}_p$, we use $\boldsymbol{A}_p \simeq \boldsymbol{B}_p$ to denote a certain asymptotic equivalence; see Appendix A.3 for a precise definition.

## A  Technical background

### A.1  Basics of free probability theory

In this section, we briefly review definitions from free probability theory and its applications to random matrices. This review will help set the stage by introducing the various mathematical structures and spaces we are working with. It will also introduce some of the notation used throughout the text.

Free probability is a mathematical framework that deals with non-commutative random variables [19]. The use of free probability theory has appeared in various recent works in statistical machine learning, including [1, 2, 11, 12, 17]. Good references on free probability theory include [3, 13], from

which we borrow some basic definitions in the following. All the material in this section is standard in free probability theory and mainly serves to keep the definitions self-contained.

**Definition 8** (Non-commutative algebra). A set $\mathcal{A}$ is called a (complex) algebra (over the field of complex numbers $\mathbb{C}$) if it is a vector space (over $\mathbb{C}$ with addition $+$), equipped with a bilinear multiplication $\cdot$, such that for all $x, y, z \in \mathcal{A}$ and $\alpha \in \mathbb{C}$,

(1) $x \cdot (y \cdot z) = (x \cdot y) \cdot z$,

(2) $(x + y) \cdot z = x \cdot z + y \cdot z$,

(3) $x \cdot (y + z) = x \cdot y + x \cdot z$,

(4) $\alpha(x \cdot y) = (\alpha x) \cdot y = x \cdot (\alpha y)$.

In addition, an algebra is called unital if a multiplicative identity element exists. We will use $1_\mathcal{A}$ to denote this identity element. We will drop the "$\cdot$" symbol to denote multiplication over the algebra.

**Definition 9** (Non-commutative probability space). Let $\mathcal{A}$ over $\mathbb{C}$ be a unital algebra with identity $1_\mathcal{A}$. Let $\varphi : \mathcal{A} \to \mathbb{C}$ be a linear functional which is unital (that is, $\varphi(1_\mathcal{A}) = 1$). Then $(\mathcal{A}, \varphi)$ is called a non-commutative probability space, and $\varphi$ is called a state. A state $\varphi$ is said to be tracial if $\varphi(xy) = \varphi(yx)$ for all $x, y \in \mathcal{A}$.

**Definition 10** (Moments). Let $(\mathcal{A}, \varphi)$ be a non-commutative probability space. The numbers $\{\varphi(x^k)\}_{k=1}^\infty$ are called the moments of the variable $x \in \mathcal{A}$.

**Definition 11** ($*$-algebra). An algebra $\mathcal{A}$ is called a $*$-algebra if there exists a mapping $x \to x^*$ from $\mathcal{A} \to \mathcal{A}$ such that, for all $x, y \in \mathcal{A}$ and $\alpha \in \mathbb{C}$,

(1) $(x + y)^* = x^* + y^*$,

(2) $(\alpha x)^* = \bar{\alpha} x^*$,

(3) $(xy)^* = y^* x^*$,

(4) $(x^*)^* = x$.

A variable $x$ of a $*$-algebra is called self-adjoint if $x = x^*$. A unital linear functional $\varphi$ on a $*$-algebra is said to be positive if $\varphi(x^* x) \geq 0$ for all $x \in \mathcal{A}$.

**Definition 12** ($*$-probability space). Let $\mathcal{A}$ be a unital $*$-algebra with a positive state $\varphi$. Then $(\mathcal{A}, \varphi)$ is called a $*$-probability space.

**Example 1.** Denote by $\mathcal{M}_p(\mathbb{C})$ the collection of all $p \times p$ matrices with complex entries. Let the multiplication and addition operations be defined in the usual way. The $*$-operation is the same as taking the conjugate transpose. Let $\mathrm{tr} : \mathcal{M}_p(\mathbb{C}) \to \mathbb{C}$ be the normalized trace defined by:

$$\overline{\mathrm{tr}}(\boldsymbol{A}) = \frac{1}{p} \mathrm{tr}[\boldsymbol{A}].$$

The state $\mathrm{tr}$ is tracial and positive.

**Definition 13** (Free independence). Suppose $(\mathcal{A}, \varphi)$ is a $*$-probability space. Then, the $*$-sub-algebras $\{\mathcal{A}_i\}_{i \in I}$ of $\mathcal{A}$ are said to be $*$-freely independent (or simply $*$-free) if, for all $n \geq 2$ and all $x_1, x_2, \cdots, x_n$ from $\{\mathcal{A}_i\}_{i \in I}$, $\kappa_n(x_1, x_2, \cdots, x_n) = 0$ whenever at least two of the $x_i$ are from different $\mathcal{A}_i$. In particular, any collection of variables is said to be $*$-free if the sub-algebras generated by these variables are $*$-free.

**Lemma 14.** Suppose $(\mathcal{A}, \varphi)$ is a $*$-probability space. If $x$ and $y$ are free in $(\mathcal{A}, \varphi)$, then for all non-negative integers $n$ and $m$,

$$\varphi(x^n y^m) = \varphi(x^n)\varphi(y^m) = \varphi(y^m x^n).$$

In other words, elements of the algebra are considered free if any alternating product of centered polynomials is also centered.

In this work, we will consider $\varphi$ to be the normalized trace. The normalized trace is the generalization of $\frac{1}{p} \mathrm{tr}[\boldsymbol{A}]$ for $\boldsymbol{A} \in \mathbb{C}^{p \times p}$ to elements of a $C^*$-algebra $\mathcal{A}$. Specifically, for any self-adjoint $a \in \mathcal{A}$ and any polynomial $p$, we have

$$\varphi(p(a)) = \int p(z) \, \mathrm{d}\mu_a(z),$$

where $\mu_a$ is the probability measure that characterizes the spectral distribution of $a$.

**Definition 15** (Convergence in spectral distribution). Let $(\mathcal{A}, \varphi)$ be a $C^*$-probability space. We say that $\boldsymbol{A}_1, \ldots, \boldsymbol{A}_m \in \mathbb{C}^{p \times p}$ *converge in spectral distribution* to elements $a_1, \ldots, a_m \in \mathcal{A}$ if, for all $1 \le \ell < \infty$ and $1 \le i_j \le m$ for $1 \le j \le \ell$, we have

$$\frac{1}{p} \operatorname{tr}[\boldsymbol{A}_{i_1} \cdots \boldsymbol{A}_{i_\ell}] \to \varphi(a_{i_1} \cdots a_{i_\ell}).$$

Then, with slight abuse of notation, two matrices $\boldsymbol{A}, \boldsymbol{B} \in \mathbb{R}^{p \times p}$ are said to be free if

$$\frac{1}{p} \operatorname{tr}\left[\prod_{\ell=1}^{L} \operatorname{poly}_\ell^{\boldsymbol{A}}(\boldsymbol{A}) \operatorname{poly}_\ell^{\boldsymbol{B}}(\boldsymbol{B})\right] = 0,$$

for all $L \ge 1$ and all centered polynomials, that is, $\overline{\operatorname{tr}}[\operatorname{poly}_\ell^{\boldsymbol{A}}(\boldsymbol{A})] = 0$. This notation is an abuse of notation because finite matrices cannot satisfy this condition. However, they can satisfy it asymptotically as $p \to \infty$, and in this case, we say that $\boldsymbol{A}$ and $\boldsymbol{B}$ are *asymptotically free*.

Note: With some abuse of notation, we will let matrices in boldface denote both the finite matrix and the limiting element in the free probability space. The limiting element can be understood, for example, as a bounded linear operator on a Hilbert space. We also remark that all notions we need are well-defined in this limit as well, as long as they are appropriately normalized.

## A.2 Useful transforms and their relationships

In this section, we review the key transforms used in free probability theory and their interrelationships.

**Definition 16** (Cauchy transform). Let $a$ be an element of a $*$-probability space $(\mathcal{A}, \varphi)$. Suppose there exists some $C > 0$ such that $|\varphi(a^n)| \le C^n$ for all $n \in \mathbb{N}$. Then the Cauchy transform of $a$ is defined as:

$$\mathcal{G}_a(z) = \sum_{n=0}^{\infty} \frac{\varphi(a^n)}{z^{n+1}}$$

for all $z \in \mathbb{C}$ with $|z| > C$.

Note that the Cauchy transform is the negative of the Stieltjes transform. In this paper, we will focus only on the Cauchy transform. Recall that for a probability measure $\nu$ on $\mathbb{R}$ and for $z \notin \mathbb{R}$, the Cauchy transform of $\nu$ is defined as:

$$\mathcal{G}(z) = \int_{\mathbb{R}} \frac{1}{z - x} \, \mathrm{d}\nu(x).$$

The definition above is motivated by the following property of the Cauchy transform of a measure. Suppose $\nu$ is a probability measure whose support is contained in $[-C, C]$ for some $C > 0$ and which has moments $\{m_k(\nu)\}_{k=0}^{\infty}$. Then the Cauchy transform of $\nu$ is defined for $z \in \mathbb{C}$ with $|z| > C$ as:

$$\mathcal{G}_\nu(z) = \sum_{k=0}^{\infty} \frac{m_k(\nu)}{z^{k+1}}.$$

**Definition 17** (Moment generating function). Let $a$ be an element of a $*$-probability space $(\mathcal{A}, \varphi)$. The moment generating function of $a$ is defined as:

$$\mathcal{M}_a(z) = 1 + \sum_{k=1}^{\infty} \varphi(a^k) z^k$$

for $z \in \mathbb{C}$ such that $|z| < r_a$. Here, $r_a$ is the radius of convergence of the series.

For a probability measure $\nu$, the moment generating function is defined analogously. (Note: The definition above is not to be confused with the moment generating function of a random variable in probability theory.) The Cauchy transform is related to the moment series via:

$$\mathcal{G}_a(z) = \frac{1}{z} \mathcal{M}_a\left(\frac{1}{z}\right). \tag{13}$$

In the other direction, we have:

$$\mathcal{M}_a(z) = \frac{1}{z}\mathcal{G}_a\left(\frac{1}{z}\right) - 1. \tag{14}$$

**Definition 18** ($S$-transform). For

$$\mathcal{M}_a(z) = \sum_{m=0}^{\infty} \varphi(a^m)z^m,$$

we define the $S$-transform of $a$ by:

$$\mathcal{S}_a(w) = \frac{1+w}{w}\mathcal{M}_a^{\langle -1 \rangle}(w), \tag{15}$$

where $\mathcal{M}^{\langle -1 \rangle}$ denotes the inverse under composition of $\mathcal{M}$.

Finally, in terms of operator $\boldsymbol{A}$, we summarize the series of invertible transformations between the various transforms introduced in this section.

- *Cauchy transform*:
$$\mathcal{G}_{\boldsymbol{A}}(z) = \overline{\mathrm{tr}}[(z\boldsymbol{I} - \boldsymbol{A})^{-1}].$$

- *Moment generating series*:
$$\mathcal{M}_{\boldsymbol{A}}(z) = \frac{1}{z}\mathcal{G}_{\boldsymbol{A}}\left(\frac{1}{z}\right) - 1.$$

- *S-transform*:
$$\mathcal{S}_{\boldsymbol{A}}(w) = \frac{1+w}{w}\mathcal{M}_{\boldsymbol{A}}^{\langle -1 \rangle}(w).$$

Here:

- $\mathcal{M}_{\boldsymbol{A}}(z) = \sum_{k=1}^{\infty} \overline{\mathrm{tr}}[\boldsymbol{A}^k]z^k$ is the moment generating series.
- $\mathcal{M}_{\boldsymbol{A}}^{\langle -1 \rangle}$ denotes the inverse under composition of $\mathcal{M}_{\boldsymbol{A}}$.
- $\overline{\mathrm{tr}}[\boldsymbol{A}]$ denotes the average trace $\mathrm{tr}[\boldsymbol{A}]/p$ of a matrix $\boldsymbol{A} \in \mathbb{R}^{p\times p}$.

### A.3 Asymptotic ridge resolvents

In this section, we provide a brief background on the language of asymptotic equivalents used in the proofs throughout the paper. We will state the definition of asymptotic equivalents and point to useful calculus rules. For more details, see [16, Appendix S.7].

To concisely present our results, we will use the framework of asymptotic equivalence [5, 6, 16], defined as follows. Let $\boldsymbol{A}_p$ and $\boldsymbol{B}_p$ be sequences of matrices of arbitrary dimensions (including vectors and scalars). We say that $\boldsymbol{A}_p$ and $\boldsymbol{B}_p$ are *asymptotically equivalent*, denoted as $\boldsymbol{A}_p \simeq \boldsymbol{B}_p$, if $\lim_{p\to\infty} |\mathrm{tr}[\boldsymbol{C}_p(\boldsymbol{A}_p - \boldsymbol{B}_p)]| = 0$ almost surely for any sequence of random matrices $\boldsymbol{C}_p$ with bounded trace norm that are independent of $\boldsymbol{A}_p$ and $\boldsymbol{B}_p$. Note that for sequences of scalar random variables, the definition simply reduces to the typical almost sure convergence of sequences of random variables involved.

The notion of deterministic equivalents obeys various calculus rules such as sum, product, differentiation, conditioning, and substitution. We refer the reader to [16] for a comprehensive list of these calculus rules, their proofs, and other related details.

Next, we collect first- and second-order asymptotic equivalents for sketched ridge resolvents from [11, 17], which will be useful for our extensions to weighted ridge resolvents.

**Assumption B** (Sketch structure). Let $\boldsymbol{S} \in \mathbb{R}^{p\times q}$ be the feature sketching matrix and $\boldsymbol{X} \in \mathbb{R}^{n\times p}$ be the data matrix. Let $\boldsymbol{S}\boldsymbol{S}^{\top}$ and $\frac{1}{n}\boldsymbol{X}^{\top}\boldsymbol{X}$ converge almost surely to bounded operators that are infinitesimally free with respect to $(\frac{1}{p}\mathrm{tr}[\cdot], \mathrm{tr}[\boldsymbol{\Theta}(\cdot)])$ for any $\boldsymbol{\Theta}$ independent of $\boldsymbol{S}$ with $\|\boldsymbol{\Theta}\|_{\mathrm{tr}}$ uniformly bounded. Additionally, let $\boldsymbol{S}\boldsymbol{S}^{\top}$ have a limiting $S$-transform that is analytic on the lower half of the complex plane.

For the statement to follow, let us define $\widehat{\mathbf{\Sigma}} := \frac{1}{n}\mathbf{X}^\top\mathbf{X}$. Let $\widetilde{\lambda}_0 := -\liminf_{p\to\infty}\lambda_{\min}^+(\mathbf{S}^\top\widehat{\mathbf{\Sigma}}\mathbf{S})$. Here, recall that $\lambda_{\min}^+(\mathbf{A})$ represents the minimum nonzero eigenvalue of a symmetric matrix $\mathbf{A}$.

**Theorem 19** (Free sketching equivalence; [11], Theorem 7.2). *Under Assumption B, for all $\lambda > \widetilde{\lambda}_0$,*

$$\mathbf{S}(\mathbf{S}^\top\widehat{\mathbf{\Sigma}}\mathbf{S} + \lambda\mathbf{I}_q)^\dagger\mathbf{S}^\top \simeq (\widehat{\mathbf{\Sigma}} + \nu\mathbf{I}_p)^\dagger, \tag{16}$$

*where $\nu > -\lambda_{\min}^+(\widehat{\mathbf{\Sigma}})$ is increasing in $\lambda > \widetilde{\lambda}_0$ and satisfies:*

$$\nu \simeq \lambda\mathcal{S}_{\mathbf{S}\mathbf{S}^\top}(-\overline{\mathrm{tr}}[\widehat{\mathbf{\Sigma}}\mathbf{S}(\mathbf{S}^\top\widehat{\mathbf{\Sigma}}\mathbf{S} + \lambda\mathbf{I}_q)^\dagger\mathbf{S}^\top]) \simeq \lambda\mathcal{S}_{\mathbf{S}\mathbf{S}^\top}(-\overline{\mathrm{tr}}[\widehat{\mathbf{\Sigma}}(\widehat{\mathbf{\Sigma}} + \nu\mathbf{I}_p)^\dagger]). \tag{17}$$

**Lemma 20** (Second-order equivalence for sketched ridge resolvents; [17], Lemma 15). Under the settings of Lemma 21, for any positive semidefinite $\mathbf{\Psi}$ with uniformly bounded operator norm, for all $\lambda > \widetilde{\lambda}_0$,

$$\mathbf{S}(\mathbf{S}^\top\widehat{\mathbf{\Sigma}}\mathbf{S} + \lambda\mathbf{I}_q)^\dagger\mathbf{S}^\top\mathbf{\Psi}\mathbf{S}(\mathbf{S}^\top\widehat{\mathbf{\Sigma}}\mathbf{S} + \lambda\mathbf{I}_q)^\dagger\mathbf{S}^\top \simeq (\widehat{\mathbf{\Sigma}} + \nu\mathbf{I}_p)^\dagger(\mathbf{\Psi} + \nu'_{\mathbf{\Psi}}\mathbf{I}_p)(\widehat{\mathbf{\Sigma}} + \nu\mathbf{I}_p)^\dagger, \tag{18}$$

where $\nu'_{\mathbf{\Psi}} \geq 0$ is given by:

$$\nu'_{\mathbf{\Psi}} = -\frac{\partial\nu}{\partial\lambda}\lambda^2\mathcal{S}'_{\mathbf{S}\mathbf{S}^\top}(-\overline{\mathrm{tr}}[\widehat{\mathbf{\Sigma}}(\widehat{\mathbf{\Sigma}} + \nu\mathbf{I}_p)^\dagger])\,\overline{\mathrm{tr}}[(\widehat{\mathbf{\Sigma}} + \nu\mathbf{I}_p)^\dagger\mathbf{\Psi}(\widehat{\mathbf{\Sigma}} + \nu\mathbf{I}_p)^\dagger]. \tag{19}$$

# B  Proofs in Section 3

## B.1  Proof of Theorem 1

Our main ingredient in the proof is Lemma 21. We will first show estimator equivalence and then show degrees of freedom equivalence.

*Estimator equivalence.* Recall from (1) the ridge estimator on the weighted data is:

$$\widehat{\boldsymbol{\beta}}_{\mathbf{W},\lambda} = (\mathbf{\Phi}^\top\mathbf{W}^\top\mathbf{W}\mathbf{\Phi}/n + \lambda\mathbf{I}_p)^\dagger\mathbf{\Phi}^\top\mathbf{W}^\top\mathbf{W}\mathbf{y}/n.$$

This is the "primal" form of the ridge estimator. Using the Woodbury matrix identity, we first write the estimator into its "dual" form.

$$\begin{aligned}\widehat{\boldsymbol{\beta}}_{\mathbf{W},\lambda} &= \mathbf{\Phi}^\top\mathbf{W}^\top(\mathbf{W}\mathbf{\Phi}\mathbf{\Phi}^\top\mathbf{W}^\top/n + \lambda\mathbf{I}_n)^\dagger\mathbf{W}\mathbf{y}/n\\ &= \mathbf{\Phi}^\top\mathbf{W}^\top(\mathbf{G}_{\mathbf{W}} + \lambda\mathbf{I}_n)^\dagger\mathbf{W}\mathbf{y}/n.\end{aligned}$$

Now, we can apply the first part of Lemma 21 to the matrix $\mathbf{W}^\top(\mathbf{G}_{\mathbf{W}} + \lambda\mathbf{I}_n)^\dagger\mathbf{W}$. From (31), we then have the following equivalence:

$$\begin{aligned}\widehat{\boldsymbol{\beta}}_{\mathbf{W},\lambda} &\simeq \mathbf{\Phi}^\top(\mathbf{G}_{\mathbf{I}} + \mu\mathbf{I}_n)^\dagger\mathbf{y}/n\\ &= \mathbf{\Phi}^\top(\mathbf{\Phi}\mathbf{\Phi}^\top + \mu\mathbf{I}_n)^\dagger\mathbf{y}/n\\ &= (\mathbf{\Phi}^\top\mathbf{\Phi}/n + \mu\mathbf{I}_n)^\dagger\mathbf{\Phi}^\top\mathbf{y}/n = \widehat{\boldsymbol{\beta}}_{\mathbf{I},\mu},\end{aligned}$$

where $\mu$ satisfies the following equation:

$$\mu = \lambda\mathcal{S}_{\mathbf{W}^\top\mathbf{W}}\left(-\frac{\mathrm{tr}[\mathbf{G}_{\mathbf{I}}(\mathbf{G}_{\mathbf{I}} + \mu\mathbf{I}_n)^\dagger]}{n}\right) = \lambda\mathcal{S}_{\mathbf{W}^\top\mathbf{W}}(-\overline{\mathrm{df}}(\widehat{\boldsymbol{\beta}}_{\mathbf{I},\mu})).$$

Note that in the simplification, we used the Woodbury identity again to go back from the dual form into the primal form for the ridge estimator based on the full data. Rearranging, we obtain the desired estimator equivalence. We next move on to showing the degrees of freedom equivalence.

*Degrees of freedom equivalence.* For the subsampled estimator $\widehat{\boldsymbol{\beta}}_{\mathbf{W},\lambda}$, the effective degrees of freedom is given by:

$$\begin{aligned}\mathrm{df}(\widehat{\boldsymbol{\beta}}_{\mathbf{W},\lambda}) &= \mathrm{tr}[\mathbf{\Phi}^\top\mathbf{\Phi}/n(\mathbf{\Phi}^\top\mathbf{\Phi}/n + \lambda\mathbf{I}_p)^\dagger]\\ &= \mathrm{tr}[\mathbf{\Phi}(\mathbf{\Phi}^\top\mathbf{\Phi}/n + \lambda\mathbf{I}_p)^\dagger\mathbf{\Phi}^\top/n]\\ &= \mathrm{tr}[(\mathbf{\Phi}\mathbf{\Phi}^\top/n + \lambda\mathbf{I}_n)^\dagger\mathbf{\Phi}\mathbf{\Phi}^\top/n].\end{aligned}$$

The second equality above follows from the push-through identity $\boldsymbol{\Phi}(\boldsymbol{\Phi}^\top\boldsymbol{\Phi}/n + \lambda\boldsymbol{I}_p)^\dagger\boldsymbol{\Phi}^\top = (\boldsymbol{\Phi}\boldsymbol{\Phi}^\top + \lambda\boldsymbol{I}_n)^\dagger\boldsymbol{\Phi}\boldsymbol{\Phi}^\top$. Recognizing the quantity inside the trace as the degrees of freedom of the full ridge estimator, we have

$$\mu = \lambda\mathcal{S}_{\boldsymbol{W}^\top\boldsymbol{W}}(-\mathsf{df}(\widehat{\boldsymbol{\beta}}_{\boldsymbol{I},\mu})).$$

We can equivalently write the equation above as

$$-\mathcal{S}_{\boldsymbol{W}^\top\boldsymbol{W}}^{-1}\left(\frac{\mu}{\lambda}\right) = \mathsf{df}(\widehat{\boldsymbol{\beta}}_{\boldsymbol{I},\mu}) \quad \text{or} \quad \frac{\mu}{\lambda} = \mathcal{S}_{\boldsymbol{W}^\top\boldsymbol{W}}(-\mathsf{df}(\widehat{\boldsymbol{\beta}}_{\boldsymbol{I},\mu})).$$

Rearranging the display above provides the desired degrees of freedom equivalence and finishes the proof.

## B.2   Proof of Theorem 2

We will apply Theorem 1 to the subsampling weight matrix $\boldsymbol{W}$. The main ingredient that we need is the $S$-transform of the spectrum of the matrix $\boldsymbol{W}^\top\boldsymbol{W}$. As summarized in Appendix A.2, one approach to compute the $S$-transform is to go through the following chain of transforms. First, we apply the Cauchy transform, then the moment-generating series, and finally, take the inverse to obtain the $S$-transform. We will do this in the following steps.

*Cauchy transform.* Recall that the Cauchy transform from Definition 16 can be computed as:

$$\mathcal{G}_{\boldsymbol{W}^\top\boldsymbol{W}}(z) = \overline{\mathrm{tr}}[(z\boldsymbol{I}_n - \boldsymbol{W}^\top\boldsymbol{W})^{-1}].$$

*Moment generating series.* We can then compute the moment series from Definition 17 using (14) as follows:

$$\begin{aligned}
\mathcal{M}_{\boldsymbol{W}^\top\boldsymbol{W}}(z) &= \frac{1}{z}\overline{\mathrm{tr}}\left[\left(\frac{1}{z}\boldsymbol{I}_n - \boldsymbol{W}^\top\boldsymbol{W}\right)^{-1}\right] - 1 \\
&= \overline{\mathrm{tr}}[(\boldsymbol{I}_n - z\boldsymbol{W}^\top\boldsymbol{W})^{-1}] - \overline{\mathrm{tr}}[\boldsymbol{I}_n] \\
&= -\overline{\mathrm{tr}}[\boldsymbol{I}_n] + \overline{\mathrm{tr}}[(\boldsymbol{I}_n - z\boldsymbol{W}^\top\boldsymbol{W})^{-1}] \\
&= \overline{\mathrm{tr}}[(z\boldsymbol{W}^\top\boldsymbol{W} - \boldsymbol{I}_n + \boldsymbol{I}_n)(\boldsymbol{I}_n - z\boldsymbol{W}^\top\boldsymbol{W})^{-1}] \\
&= \overline{\mathrm{tr}}[z\boldsymbol{W}^\top\boldsymbol{W}(\boldsymbol{I}_n - z\boldsymbol{W}^\top\boldsymbol{W})^{-1}].
\end{aligned}$$

We now note that the matrix $\boldsymbol{W}^\top\boldsymbol{W}$ has $k$ eigenvalues of 1 and $n - k$ eigenvalues of 0. Therefore, we have

$$\begin{aligned}
\mathcal{M}_{\boldsymbol{W}^\top\boldsymbol{W}}(z) &= \overline{\mathrm{tr}}[z\boldsymbol{W}^\top\boldsymbol{W}(\boldsymbol{I}_n - z\boldsymbol{W}^\top\boldsymbol{W})^{-1}] \\
&= \frac{1}{n}\left(\sum_{i=1}^n \frac{zd_i}{1 - zd_i}\right) \\
&= \frac{k}{n}\cdot\frac{z}{1 - z}.
\end{aligned} \tag{20}$$

*S-transform.* The inverse of the moment generating series map $z \mapsto \mathcal{M}_{\boldsymbol{W}^\top\boldsymbol{W}}(z)$ from (20) is:

$$\mathcal{M}^{\langle-1\rangle}(w) = \frac{w}{w + k/n}. \tag{21}$$

Therefore, from Definition 18 and using (21), we have

$$\mathcal{S}(w) = \frac{1 + w}{w}\cdot\frac{w}{w + k/n} = \frac{1 + w}{w + k/n}. \tag{22}$$

Now, we are ready to apply Theorem 1 to the subsampling matrix $\boldsymbol{W}$.

Substituting (22) into (2), we get

$$\frac{\mu}{\lambda} = \mathcal{S}(-\overline{\mathsf{df}}(\widehat{\boldsymbol{\beta}}_{\boldsymbol{I},\mu})) = \frac{1 - \overline{\mathsf{df}}(\widehat{\boldsymbol{\beta}}_{\boldsymbol{I},\mu})}{-\overline{\mathsf{df}}(\widehat{\boldsymbol{\beta}}_{\boldsymbol{I},\mu}) + k/n}.$$

Rearranging, we obtain

$$\overline{\mathsf{df}}(\widehat{\boldsymbol{\beta}}_{\boldsymbol{I},\mu}) \cdot (\mu - \lambda) = \mu \cdot (k/n) - \lambda.$$

Thus, we get

$$\overline{\mathsf{df}}(\widehat{\boldsymbol{\beta}}_{\boldsymbol{I},\mu}) = -\frac{\lambda - \mu \cdot (k/n)}{\mu - \lambda}.$$

In other words, we have

$$1 - \overline{\mathsf{df}}(\widehat{\boldsymbol{\beta}}_{\boldsymbol{I},\mu}) = \left(\frac{\mu}{\mu - \lambda}\right) \cdot \left(1 - \frac{k}{n}\right).$$

Multiplying $(1 - \lambda/\mu)$ on both sides, we arrive at the desired relation. This completes the proof.

### B.3 Proof of Proposition 3

We prove this by matching the path (4) with the one in [15]. Let $\gamma = p/n$, $\psi = p/k$, $H_p$ be the spectral distribution of $\widehat{\boldsymbol{\Sigma}} = \boldsymbol{X}^\top \boldsymbol{X}/n$. The path from Equation (5) of [15] is given by the following equation:

$$\mu = (\psi - \gamma) \int \frac{r}{1 + v(\mu, \gamma)r} \, \mathrm{d}H_p(r), \tag{23}$$

where $v(\mu, \gamma)$ is the unique solution to the following fixed-point equation:

$$\frac{1}{v(\mu, \gamma)} = \mu + \gamma \int \frac{r}{1 + v(\mu, \gamma)r} \, \mathrm{d}H_p(r) = \psi \int \frac{r}{1 + v(\mu, \gamma)r} \, \mathrm{d}H_p(r). \tag{24}$$

For given $\gamma$ and $\mu$, we will show that $\psi$ that solves (23) gives rise $k = p/\psi$ that also solves (4) with $\lambda = 0$:

$$-\frac{1}{n} \operatorname{tr}\left[\frac{1}{n}\boldsymbol{X}\boldsymbol{X}^\top \left(\frac{1}{n}\boldsymbol{X}\boldsymbol{X}^\top + \mu\boldsymbol{I}_n\right)^\dagger\right] = -\frac{k}{n}.$$

Rearranging the above equation yields:

$$\frac{k}{n} = 1 - \mu \, \overline{\operatorname{tr}}\left[\left(\frac{1}{n}\boldsymbol{X}\boldsymbol{X}^\top + \mu\boldsymbol{I}_n\right)^\dagger\right] = 1 - \mu v(\mu, \gamma),$$

where the second equality is from Lemma B.2 of [15]. This implies that

$$\begin{aligned}
\mu &= \left(1 - \frac{k}{n}\right) \frac{1}{v(\mu, \gamma)} \\
&= \left(1 - \frac{k}{n}\right) \psi \int \frac{r}{1 + v(\mu, \gamma)r} \, \mathrm{d}H_p(r) \\
&= (\psi - \gamma) \int \frac{r}{1 + v(\mu, \gamma)r} \, \mathrm{d}H_p(r),
\end{aligned}$$

where the second equality follows from (24). The above is the same as the path (23) in [15]. This finishes the proof.

### B.4 Proof of Proposition 4

We first describe the setup for the kernel ridge regression formulation and then show the desired equivalence.

*Setup.* Let $K(\cdot, \cdot) : \mathbb{R}^d \times \mathbb{R}^d \to \mathbb{R}$ be a kernel function. Let $\mathcal{H}$ denote the reproducing kernel Hilbert space associated with kernel $K$. Kernel ridge regression with the subsampling matrix $\boldsymbol{W}$ solves the following problem with tuning parameter $\lambda \geq 0$:

$$\widehat{f}_{\boldsymbol{W},\lambda} = \underset{f \in \mathcal{H}}{\operatorname{argmin}} \, \|\boldsymbol{W}\boldsymbol{y} - \boldsymbol{W}f(\boldsymbol{X})\|_2^2/n + \lambda\|f\|_{\mathcal{H}}^2,$$

where $f(\boldsymbol{X}) = [f(\boldsymbol{x}_1), \dots, f(\boldsymbol{x}_n)]^\top$. Kernel ridge regression predictions have a closed-form expression:

$$\widehat{f}_{\boldsymbol{W},\lambda}(\boldsymbol{x}) = K(\boldsymbol{x}, \boldsymbol{X})^\top \boldsymbol{W}^\top (\boldsymbol{W}K(\boldsymbol{X}, \boldsymbol{X})\boldsymbol{W}^\top + \lambda\boldsymbol{I}_n)^\dagger \boldsymbol{W}\boldsymbol{y}.$$

Here, $K(\boldsymbol{x}, \boldsymbol{X}) \in \mathbb{R}^n$ with $i$-th entry $K(\boldsymbol{x}, \boldsymbol{x}_i)$, and $K(\boldsymbol{X}, \boldsymbol{X}) \in \mathbb{R}^{n \times n}$ with the $ij$-th entry $K(\boldsymbol{x}_i, \boldsymbol{x}_j)$.

The predicted values on the training data $\boldsymbol{X}$ are given by

$$\widehat{f}_{\boldsymbol{W}, \lambda}(\boldsymbol{X}) = K(\boldsymbol{X}, \boldsymbol{X})^\top \boldsymbol{W}^\top (\boldsymbol{W} K(\boldsymbol{X}, \boldsymbol{X}) \boldsymbol{W}^\top + \lambda \boldsymbol{I}_n)^\dagger \boldsymbol{W} \boldsymbol{y}.$$

Here, the matrix $K(\boldsymbol{X}, \boldsymbol{X})^\top \boldsymbol{W}^\top (\boldsymbol{W} K(\boldsymbol{X}, \boldsymbol{X}) \boldsymbol{W}^\top + \lambda \boldsymbol{I}_n)^\dagger \boldsymbol{W}$ is the smoothing matrix.

Define $\boldsymbol{G_I} = K(\boldsymbol{X}, \boldsymbol{X})$ and $\boldsymbol{G_W} = \boldsymbol{W} K(\boldsymbol{X}, \boldsymbol{X}) \boldsymbol{W}^\top$. Leveraging the kernel trick, the preceding optimization problem translates into solving the following problem (in the dual domain):

$$\widehat{\boldsymbol{\alpha}}_{\boldsymbol{W}, \lambda} = \underset{\boldsymbol{\alpha} \in \mathbb{R}^n}{\arg\min} \, \boldsymbol{\alpha}^\top (\boldsymbol{G_W} + \lambda \boldsymbol{I}_n) \boldsymbol{\alpha} + 2 \boldsymbol{\alpha}^\top \boldsymbol{W} \boldsymbol{y},$$

where the dual solution is given by $\widehat{\boldsymbol{\alpha}}_{\boldsymbol{W}, \lambda} = (\boldsymbol{G_W} + \lambda \boldsymbol{I}_n)^\dagger \boldsymbol{W} \boldsymbol{y}$. The correspondence between the dual and primal solutions is simply given by: $\widehat{\boldsymbol{\beta}}_{\boldsymbol{W}, \lambda} = \boldsymbol{\Phi}^\top \boldsymbol{W}^\top \widehat{\boldsymbol{\alpha}}_{\boldsymbol{W}, \lambda}$ where $\boldsymbol{\Phi} = [\phi(\boldsymbol{x}_1), \dots, \phi(\boldsymbol{x}_n)]^\top$ is the feature matrix and $\phi \colon \mathbb{R}^d \mapsto \mathcal{H}$ is the feature map of the Hilbert space $\mathcal{H}$ with kernel $K$. Thus, $\widehat{f}_{\boldsymbol{W}, \lambda}(\boldsymbol{X}) = \boldsymbol{W} \boldsymbol{\Phi} \widehat{\boldsymbol{\beta}}_{\boldsymbol{W}, \lambda} = \boldsymbol{W} \boldsymbol{\Phi} \boldsymbol{\Phi}^\top \boldsymbol{W}^\top \widehat{\boldsymbol{\alpha}}_{\boldsymbol{W}, \lambda} = \boldsymbol{G_W} (\boldsymbol{G_W} + \lambda \boldsymbol{I}_n)^\dagger \boldsymbol{W} \boldsymbol{y}$ and the degrees of freedom is given by $\mathrm{df}(\widehat{\boldsymbol{\beta}}_{\boldsymbol{I}, \mu}) = \mathrm{tr}[\boldsymbol{G_W} (\boldsymbol{G_W} + \lambda \boldsymbol{I}_n)^\dagger]$.

Next, we show that (3) holds. Alternatively, one can also show that

$$\widehat{\boldsymbol{\alpha}}_{\boldsymbol{W}, \lambda} \simeq \widehat{\boldsymbol{\alpha}}_{\boldsymbol{I}, \mu}, \quad \text{and} \quad \widehat{f}_{\boldsymbol{W}, \lambda}(\boldsymbol{x}_0) \simeq \widehat{f}_{\boldsymbol{I}, \mu}(\boldsymbol{x}_0),$$

which we omit due to similarity. Our proof strategy consists of two steps. We first show that it suffices to establish the desired result for the linearized version. We then show that we can suitably adapt our result for the linearized version.

*Linearization of kernels.* In the below, we will show that for $\mu \geq 0$,

$$\boldsymbol{W}^\top (\boldsymbol{G_W} + \lambda \boldsymbol{I}_n)^\dagger \boldsymbol{W} \simeq (\boldsymbol{G_I} + \mu \boldsymbol{I}_n)^\dagger,$$
$$\overline{\mathrm{tr}}[\lambda (\boldsymbol{G_W} + \lambda \boldsymbol{I}_n)^\dagger] \simeq \overline{\mathrm{tr}}[\mu (\boldsymbol{G_I} + \mu \boldsymbol{I}_n)^\dagger],$$

where $\boldsymbol{W}$ and $\lambda \geq 0$ satisfy that

$$\mu = \lambda \mathcal{S}_{\boldsymbol{W}^\top \boldsymbol{W}} (-\tfrac{1}{n} \mathrm{tr}[\boldsymbol{G_I} (\boldsymbol{G_I} + \lambda \boldsymbol{I}_n)^\dagger]) = \lambda \mathcal{S}_{\boldsymbol{W}^\top \boldsymbol{W}} (-\tfrac{1}{n} \mathrm{tr}[\boldsymbol{G_W} (\boldsymbol{G_W} + \mu \boldsymbol{I}_n)^\dagger]). \quad (25)$$

Using assumptions of Proposition 3 and the assumption in Proposition 4, by [18, Proposition 5.1],[2]

$$\|\boldsymbol{G_I} - \boldsymbol{G_I}^{\mathrm{lin}}\|_{\mathrm{op}} \xrightarrow{\mathrm{p}} 0,$$

where

$$\boldsymbol{G_I}^{\mathrm{lin}} = c_0 \boldsymbol{I}_n + c_1 \boldsymbol{1}_n \boldsymbol{1}_n^\top + c_2 \boldsymbol{X} \boldsymbol{X}^\top$$

and $(c_0, c_1, c_2)$ associated with function $g$ in Proposition 4 and $\tau = \lim_{p \to \infty} \mathrm{tr}[\boldsymbol{\Sigma}]/p$ are defined as

$$c_0 = g(\tau, \tau, \tau) - g(\tau, 0, \tau) - c_2 \frac{\mathrm{tr}[\boldsymbol{\Sigma}]}{p}, \quad (26)$$

$$c_1 = g(\tau, 0, \tau) + g''(\tau, 0, \tau) \frac{\mathrm{tr}[\boldsymbol{\Sigma}^2]}{2p^2}, \quad (27)$$

$$c_2 = g'(\tau, 0, \tau). \quad (28)$$

Assume $\boldsymbol{C}_n$ is a sequence of random matrices with bounded trace norm. Note that

$$\mathrm{tr}[\boldsymbol{C}_p ((\boldsymbol{G_I} + \mu \boldsymbol{I}_n)^\dagger - (\boldsymbol{G_I}^{\mathrm{lin}} + \mu \boldsymbol{I}_n)^\dagger)]$$
$$\leq \mathrm{tr}[\boldsymbol{C}_p] \|(\boldsymbol{G_I} + \mu \boldsymbol{I}_n)^\dagger - (\boldsymbol{G_I}^{\mathrm{lin}} + \mu \boldsymbol{I}_n)^\dagger\|_{\mathrm{op}}$$
$$\leq \mathrm{tr}[\boldsymbol{C}_p] \|(\boldsymbol{G_I} + \mu \boldsymbol{I}_n)^\dagger\|_{\mathrm{op}} \|(\boldsymbol{G_I}^{\mathrm{lin}} + \mu \boldsymbol{I}_n)^\dagger\|_{\mathrm{op}} \|\boldsymbol{G_I} - \boldsymbol{G_I}^{\mathrm{lin}}\|_{\mathrm{op}} \xrightarrow{\mathrm{a.s.}} 0.$$

---

[2] Assumption A1 of [18] requires finite $5 + \delta$-moments, which can be relaxed to only finite $4 + \delta$-moments as in the assumption of Proposition 3, by a truncation argument as in the proof of Theorem 6 of [7, Appendix A.4].

where in the last inequality, we use a matrix identity $\boldsymbol{A}^{-1} - \boldsymbol{B}^{-1} = \boldsymbol{A}^{-1}(\boldsymbol{B} - \boldsymbol{A})\boldsymbol{B}^{-1}$ for two invertible matrices $\boldsymbol{A}$ and $\boldsymbol{B}$. Thus, we have

$$(\boldsymbol{G_I} + \mu\boldsymbol{I}_n)^\dagger \simeq_p (\boldsymbol{G}_I^{\text{lin}} + \mu\boldsymbol{I}_n)^\dagger. \tag{29}$$

Hence, combining (29) and the transition property of asymptotic equivalence [15, Lemma S.7.4 (1)], it suffices to show

$$\boldsymbol{W}^\top(\boldsymbol{G}_W^{\text{lin}} + \lambda\boldsymbol{I}_n)^\dagger\boldsymbol{W} \simeq (\boldsymbol{G}_I^{\text{lin}} + \mu\boldsymbol{I}_n)^\dagger,$$

where $\boldsymbol{G}_W^{\text{lin}} = \boldsymbol{W}\boldsymbol{G}_I^{\text{lin}}\boldsymbol{W}^\top$, and $\lambda$ and $\mu$ satisfy (25). Similarly, we can also show that the path (25) is asymptotically equivalent to

$$\mu = \lambda\mathcal{S}_{\boldsymbol{W}^\top\boldsymbol{W}}(-\tfrac{1}{n}\operatorname{tr}[\boldsymbol{G}_I^{\text{lin}}(\boldsymbol{G}_I^{\text{lin}} + \lambda\boldsymbol{I}_n)^\dagger]) = \lambda\mathcal{S}_{\boldsymbol{W}^\top\boldsymbol{W}}(-\tfrac{1}{n}\operatorname{tr}[\boldsymbol{G}_W^{\text{lin}}(\boldsymbol{G}_W^{\text{lin}} + \mu\boldsymbol{I}_n)^\dagger]). \tag{30}$$

*Equivalence for linearized kernels.* We next show that the resolvent equivalence result holds for $\boldsymbol{K}^{\text{lin}}$. This follows from additional manipulations building on Lemma 21.

## B.5  Proof of Proposition 5

In the below, we will show that for $\mu \geq 0$,

$$\boldsymbol{W}^\top(\boldsymbol{W}\boldsymbol{\Phi}\boldsymbol{\Phi}^\top\boldsymbol{W}^\top/n + \lambda\boldsymbol{I}_n)^\dagger\boldsymbol{W} \simeq (\boldsymbol{\Phi}\boldsymbol{\Phi}^\top/n + \mu\boldsymbol{I}_n)^\dagger,$$
$$\overline{\operatorname{tr}}[\lambda(\boldsymbol{W}\boldsymbol{\Phi}\boldsymbol{\Phi}^\top\boldsymbol{W}^\top/n + \lambda\boldsymbol{I}_n)^\dagger] \simeq \overline{\operatorname{tr}}[\mu(\boldsymbol{\Phi}\boldsymbol{\Phi}^\top/n + \mu\boldsymbol{I}_n)^\dagger].$$

Under assumptions in Proposition 5, the linearized features take the form

$$\boldsymbol{\Phi}^{\text{lin}} = \sqrt{\frac{\rho_s}{d}}\boldsymbol{F}\boldsymbol{X} + \sqrt{\rho_s\omega_s}\boldsymbol{U},$$

where the constants $\rho_s$ and $\omega_s$ are given in Proposition 5 and $\boldsymbol{U} \in \mathbb{R}^{n \times p}$ has i.i.d. standard normal entries. From Claim A.13 of [10], the linear functionals of the estimators $\widehat{\boldsymbol{\beta}}_{\boldsymbol{D},\lambda}$ and $\widehat{\boldsymbol{\beta}}_{\boldsymbol{I},\mu}$ with random features $\boldsymbol{\Phi}$ and $\boldsymbol{\Phi}^{\text{lin}}$ are asymptotically equivalent. Now, following the proof of Proposition 4, we apply Lemma 21 on $\boldsymbol{\Phi}^{\text{lin}}$ to yield the desired result.

## B.6  Technical lemmas

In preparation for the forthcoming statement, define $\lambda_0 = -\liminf_{n\to\infty}\lambda_{\min}^+(\boldsymbol{G_W})$. Recall the Gram matrices $\boldsymbol{G} = \boldsymbol{\Phi}\boldsymbol{\Phi}^\top/n$ and $\boldsymbol{G_W} = \boldsymbol{W}\boldsymbol{\Phi}\boldsymbol{\Phi}^\top\boldsymbol{W}^\top/n$.

**Lemma 21** (General first-order equivalence for freely subsampled ridge resolvents). For $\boldsymbol{W} \in \mathbb{R}^{n \times n}$, suppose Assumption A holds for $\boldsymbol{W}\boldsymbol{W}^\top$. Then, for all $\lambda > \lambda_0$,

$$\boldsymbol{W}^\top(\boldsymbol{G_W} + \lambda\boldsymbol{I})^\dagger\boldsymbol{W} \simeq (\boldsymbol{G} + \mu\boldsymbol{I})^\dagger, \tag{31}$$

$$\overline{\operatorname{tr}}[\lambda(\boldsymbol{G_W} + \lambda\boldsymbol{I}_n)^\dagger] \simeq \overline{\operatorname{tr}}[\mu(\boldsymbol{G} + \mu\boldsymbol{I}_n)^\dagger], \tag{32}$$

where $\mu > -\lambda_{\min}^+(\boldsymbol{G})$ solves the equation:

$$\mu = \lambda\mathcal{S}_{\boldsymbol{W}\boldsymbol{W}^\top}(-\overline{\operatorname{tr}}[\boldsymbol{G}(\boldsymbol{G} + \mu\boldsymbol{V})^\dagger]) \simeq \lambda\mathcal{S}_{\boldsymbol{W}\boldsymbol{W}^\top}(-\overline{\operatorname{tr}}[\boldsymbol{G_W}(\boldsymbol{G_W} + \lambda\boldsymbol{V})^\dagger]). \tag{33}$$

*Proof of Lemma 21.* The first result follows from using Theorem 19 by suitably changing the roles of $\boldsymbol{X}$ and $\boldsymbol{\Phi}$. In particular, we set $\boldsymbol{\Phi}$ to be $\boldsymbol{X}^\top$ and $\boldsymbol{W}$ to be $\boldsymbol{S}$ and apply Theorem 19 to obtain

$$\boldsymbol{W}^\top(\boldsymbol{W}\boldsymbol{\Phi}\boldsymbol{\Phi}^\top\boldsymbol{W}^\top/n + \lambda\boldsymbol{I}_n)^\dagger\boldsymbol{W} \simeq (\boldsymbol{\Phi}\boldsymbol{\Phi}^\top/n + \mu\boldsymbol{I}_n)^\dagger. \tag{34}$$

Writing in terms of $\boldsymbol{G}$ and $\boldsymbol{G_W}$, this proves the first part (31).

For the second part, we use the result (34) in the first part and multiply both sides by $\boldsymbol{\Phi}\boldsymbol{\Phi}^\top/n$ to get

$$(\boldsymbol{\Phi}\boldsymbol{\Phi}^\top/n) \cdot \boldsymbol{W}^\top(\boldsymbol{W}\boldsymbol{\Phi}\boldsymbol{\Phi}^\top\boldsymbol{W}^\top/n + \lambda\boldsymbol{I}_n)^\dagger\boldsymbol{W} \simeq (\boldsymbol{\Phi}\boldsymbol{\Phi}^\top/n) \cdot (\boldsymbol{\Phi}\boldsymbol{\Phi}^\top/n + \mu\boldsymbol{I}_n)^\dagger.$$

Using the trace property of asymptotic equivalence [15, Lemma S.7.4 (4)], we have

$$\overline{\operatorname{tr}}[(\boldsymbol{\Phi}\boldsymbol{\Phi}^\top/n) \cdot \boldsymbol{W}^\top(\boldsymbol{W}\boldsymbol{\Phi}\boldsymbol{\Phi}^\top\boldsymbol{W}^\top/n + \lambda\boldsymbol{I}_n)^\dagger\boldsymbol{W}] \simeq \overline{\operatorname{tr}}[(\boldsymbol{\Phi}\boldsymbol{\Phi}^\top/n) \cdot (\boldsymbol{\Phi}\boldsymbol{\Phi}^\top/n + \mu\boldsymbol{I}_n)^\dagger].$$

Using the cyclic property of the trace operator yields

$$\overline{\mathrm{tr}}[(\boldsymbol{W}\boldsymbol{\Phi}\boldsymbol{\Phi}^\top/n)\cdot\boldsymbol{W}^\top(\boldsymbol{W}\boldsymbol{\Phi}\boldsymbol{\Phi}^\top\boldsymbol{W}^\top/n+\lambda\boldsymbol{I}_n)^\dagger]\simeq\overline{\mathrm{tr}}[(\boldsymbol{\Phi}\boldsymbol{\Phi}^\top/n)\cdot(\boldsymbol{\Phi}\boldsymbol{\Phi}^\top/n+\mu\boldsymbol{I}_n)^\dagger].$$

In terms of $\boldsymbol{G}$ and $\boldsymbol{G}_{\boldsymbol{W}}$, this is the same as

$$\overline{\mathrm{tr}}[\boldsymbol{G}_{\boldsymbol{W}}(\boldsymbol{G}_{\boldsymbol{W}}+\lambda\boldsymbol{I}_n)^\dagger]\simeq\overline{\mathrm{tr}}[\boldsymbol{G}(\boldsymbol{G}+\mu\boldsymbol{I}_n)^\dagger].$$

Adding and subtracting $\lambda\boldsymbol{I}_n$ and $\mu\boldsymbol{I}_n$ on the left- and right-hand resolvents, we arrive at the second part (32). This completes the proof. $\qquad\square$

## C   Proofs in Section 4

### C.1   Proof of Theorem 6

The main ingredients of the proof are Lemmas 21 and 22. We begin by decomposing the unknown response $y_0$ into its linear predictor and residual. Specifically, let $\boldsymbol{\beta}_0$ be the optimal projection parameter given by $\boldsymbol{\beta}_0 = \boldsymbol{\Sigma}_0^{-1}\mathbb{E}[\boldsymbol{\phi}_0 y_0]$. Then, we can express the response as the sum of its best linear predictor, $\boldsymbol{\phi}_0^\top\boldsymbol{\beta}_0$, and the residual, $y_0 - \boldsymbol{\phi}_0^\top\boldsymbol{\beta}_0$. Denote the variance of this residual by $\sigma_0^2 = \mathbb{E}[(y_0 - \boldsymbol{\phi}_0^\top\boldsymbol{\beta}_0)^2]$. It is easy to see that the risk decomposes as follows:

$$\begin{aligned}
R(\widehat{\boldsymbol{\beta}}_{\boldsymbol{W}_{1:M},\lambda}) &= \mathbb{E}\big[(y_0 - \boldsymbol{\phi}_0^\top\widehat{\boldsymbol{\beta}}_{\boldsymbol{W}_{1:M},\lambda})^2 \mid \boldsymbol{\Phi}, \boldsymbol{y}, \{\boldsymbol{W}_m\}_{m=1}^M\big]\\
&= (\widehat{\boldsymbol{\beta}}_{\boldsymbol{W}_{1:M},\lambda} - \boldsymbol{\beta}_0)^\top\boldsymbol{\Sigma}(\widehat{\boldsymbol{\beta}}_{\boldsymbol{W}_{1:M},\lambda} - \boldsymbol{\beta}_0) + \sigma_0^2.
\end{aligned}$$

Here, we used the fact that $(y_0 - \boldsymbol{\phi}_0^\top\boldsymbol{\beta}_0)$ is uncorrelated with $\boldsymbol{\phi}_0$, that is, $\mathbb{E}[\boldsymbol{\phi}_0(y_0 - \boldsymbol{\phi}_0^\top\boldsymbol{\beta}_0)] = \boldsymbol{0}_p$. We note that $\|\boldsymbol{\beta}_0\|_2 < \infty$ and $\boldsymbol{\Sigma}_0$ has uniformly bounded operator norm.

Observe that

$$\begin{aligned}
R(\widehat{\boldsymbol{\beta}}_{\boldsymbol{W}_{1:M},\lambda}) &= (\widehat{\boldsymbol{\beta}}_{\boldsymbol{W}_{1:M},\lambda} - \boldsymbol{\beta}_0)^\top\boldsymbol{\Sigma}_0(\widehat{\boldsymbol{\beta}}_{\boldsymbol{W}_{1:M},\lambda} - \boldsymbol{\beta}_0) + \sigma_0^2\\
&= \left(\frac{1}{M}\sum_{m=1}^M\widehat{\boldsymbol{\beta}}_{\boldsymbol{W}_m,\lambda} - \boldsymbol{\beta}_0\right)^\top\boldsymbol{\Sigma}_0\left(\frac{1}{M}\sum_{m=1}^M\widehat{\boldsymbol{\beta}}_{\boldsymbol{W}_m,\lambda} - \boldsymbol{\beta}_0\right) + \sigma_0^2\\
&= \frac{1}{M^2}\sum_{k,\ell=1}^M\widehat{\boldsymbol{\beta}}_{\boldsymbol{W}_k,\lambda}^\top\boldsymbol{\Sigma}_0\widehat{\boldsymbol{\beta}}_{\boldsymbol{W}_\ell,\lambda} - \frac{2}{M}\sum_{m=1}^M\boldsymbol{\beta}_0^\top\boldsymbol{\Sigma}_0\widehat{\boldsymbol{\beta}}_{\boldsymbol{W}_m,\lambda} + \boldsymbol{\beta}_0^\top\boldsymbol{\Sigma}_0\boldsymbol{\beta}_0 + \sigma_0^2\\
&= \frac{1}{M^2}\sum_{k,\ell=1}^M(\widehat{\boldsymbol{\beta}}_{\boldsymbol{W}_k,\lambda}^\top\boldsymbol{\Sigma}_0\widehat{\boldsymbol{\beta}}_{\boldsymbol{W}_\ell,\lambda} - \widehat{\boldsymbol{\beta}}_{\boldsymbol{I},\mu}^\top\boldsymbol{\Sigma}_0\widehat{\boldsymbol{\beta}}_{\boldsymbol{I},\mu}) + \widehat{\boldsymbol{\beta}}_{\boldsymbol{I},\mu}^\top\boldsymbol{\Sigma}_0\widehat{\boldsymbol{\beta}}_{\boldsymbol{I},\mu}\\
&\quad - \frac{2}{M}\sum_{m=1}^M\boldsymbol{\beta}_0^\top\boldsymbol{\Sigma}_0\widehat{\boldsymbol{\beta}}_{\boldsymbol{W}_m,\lambda} + \boldsymbol{\beta}_0^\top\boldsymbol{\Sigma}_0\boldsymbol{\beta}_0 + \sigma_0^2.
\end{aligned}$$

By Lemma 21, note that

$$\frac{1}{M}\sum_{k=1}^M\widehat{\boldsymbol{\beta}}_{\boldsymbol{W}_m,\lambda}\simeq\widehat{\boldsymbol{\beta}}_{\boldsymbol{I},\mu}.$$

Thus, we have

$$\begin{aligned}
R(\widehat{\boldsymbol{\beta}}_{\boldsymbol{W}_{1:M},\lambda}) &\simeq \frac{1}{M^2}\sum_{k,\ell=1}^M(\widehat{\boldsymbol{\beta}}_{\boldsymbol{W}_k,\lambda}^\top\boldsymbol{\Sigma}_0\widehat{\boldsymbol{\beta}}_{\boldsymbol{W}_\ell,\lambda} - \widehat{\boldsymbol{\beta}}_{\boldsymbol{I},\mu}^\top\boldsymbol{\Sigma}_0\widehat{\boldsymbol{\beta}}_{\boldsymbol{I},\mu})\\
&\quad + \widehat{\boldsymbol{\beta}}_{\boldsymbol{I},\mu}^\top\boldsymbol{\Sigma}_0\widehat{\boldsymbol{\beta}}_{\boldsymbol{I},\mu} - \frac{2}{M}\sum_{m=1}^M\boldsymbol{\beta}_0^\top\boldsymbol{\Sigma}_0\widehat{\boldsymbol{\beta}}_{\boldsymbol{I},\mu} + \boldsymbol{\beta}_0^\top\boldsymbol{\Sigma}_0\boldsymbol{\beta}_0 + \sigma_0^2.
\end{aligned}$$

Now, by two applications of Lemma 21, we know that $\widehat{\boldsymbol{\beta}}_{\boldsymbol{W}_k,\lambda}^\top\boldsymbol{\Sigma}_0\widehat{\boldsymbol{\beta}}_{\boldsymbol{W}_\ell,\lambda} - \widehat{\boldsymbol{\beta}}_{\boldsymbol{I},\mu}^\top\boldsymbol{\Sigma}_0\widehat{\boldsymbol{\beta}}_{\boldsymbol{I},\mu} \xrightarrow{\text{a.s.}} 0$ when $k \neq \ell$ since $\boldsymbol{W}_k$ and $\boldsymbol{W}_\ell$ are independent. Hence, we have

$$R(\widehat{\boldsymbol{\beta}}_{\boldsymbol{W}_{1:M},\lambda}) \simeq \frac{1}{M^2}\sum_{m=1}^M(\widehat{\boldsymbol{\beta}}_{\boldsymbol{W}_m,\lambda}^\top\boldsymbol{\Sigma}_0\widehat{\boldsymbol{\beta}}_{\boldsymbol{W}_m,\lambda} - \widehat{\boldsymbol{\beta}}_{\boldsymbol{I},\mu}^\top\boldsymbol{\Sigma}_0\widehat{\boldsymbol{\beta}}_{\boldsymbol{I},\mu}) + (\widehat{\boldsymbol{\beta}}_{\boldsymbol{I},\mu} - \boldsymbol{\beta}_0)^\top\boldsymbol{\Sigma}_0(\widehat{\boldsymbol{\beta}}_{\boldsymbol{I},\mu} - \boldsymbol{\beta}_0) + \sigma_0^2$$

$$\simeq \frac{1}{M}\big(\widehat{\boldsymbol{\beta}}_{\boldsymbol{W},\lambda}^\top \boldsymbol{\Sigma}_0 \widehat{\boldsymbol{\beta}}_{\boldsymbol{W},\lambda} - \widehat{\boldsymbol{\beta}}_{\boldsymbol{I},\mu}^\top \boldsymbol{\Sigma}_0 \widehat{\boldsymbol{\beta}}_{\boldsymbol{I},\mu}\big) + (\widehat{\boldsymbol{\beta}}_{\boldsymbol{I},\mu} - \boldsymbol{\beta}_0)^\top \boldsymbol{\Sigma}_0 (\widehat{\boldsymbol{\beta}}_{\boldsymbol{I},\mu} - \boldsymbol{\beta}_0) + \sigma_0^2$$

$$= \frac{1}{M}\big(\widehat{\boldsymbol{\beta}}_{\boldsymbol{W},\lambda}^\top \boldsymbol{\Sigma}_0 \widehat{\boldsymbol{\beta}}_{\boldsymbol{W},\lambda} - \widehat{\boldsymbol{\beta}}_{\boldsymbol{I},\mu}^\top \boldsymbol{\Sigma}_0 \widehat{\boldsymbol{\beta}}_{\boldsymbol{I},\mu}\big) + R(\widehat{\boldsymbol{\beta}}_{\boldsymbol{I},\mu}) \tag{35}$$

where we used the fact that the $M$ terms where $k = \ell$ converge identically in the second to last line and a risk decomposition similar to that for $\widehat{\boldsymbol{\beta}}_{\boldsymbol{W}_{1:M},\lambda}$ in the last line. Thus, it suffices to evaluate the difference $\widehat{\boldsymbol{\beta}}_\lambda^\top \boldsymbol{\Sigma}_0 \widehat{\boldsymbol{\beta}}_\lambda - \widehat{\boldsymbol{\beta}}_\mu^\top \boldsymbol{\Sigma}_0 \widehat{\boldsymbol{\beta}}_\mu$ to finish the proof.

We have

$$\widehat{\boldsymbol{\beta}}_{\boldsymbol{W},\lambda}^\top \boldsymbol{\Sigma}_0 \widehat{\boldsymbol{\beta}}_{\boldsymbol{W},\lambda} - \widehat{\boldsymbol{\beta}}_{\boldsymbol{I},\mu}^\top \boldsymbol{\Sigma}_0 \widehat{\boldsymbol{\beta}}_{\boldsymbol{I},\mu}$$

$$= (\boldsymbol{y}^\top \boldsymbol{W}^\top/n)\boldsymbol{W}\boldsymbol{\Phi}(\boldsymbol{\Phi}^\top \boldsymbol{W}^\top \boldsymbol{W}\boldsymbol{\Phi}/n + \lambda\boldsymbol{I}_p)^\dagger \boldsymbol{\Sigma}_0(\boldsymbol{\Phi}^\top \boldsymbol{W}^\top \boldsymbol{W}\boldsymbol{\Phi}/n + \lambda\boldsymbol{I}_p)^\dagger \boldsymbol{\Phi}^\top \boldsymbol{W}^\top(\boldsymbol{W}\boldsymbol{y}/n)$$

$$\quad - (\boldsymbol{y}^\top \boldsymbol{\Phi}/n)(\boldsymbol{\Phi}^\top \boldsymbol{\Phi}/n + \mu\boldsymbol{I}_p)^\dagger \boldsymbol{\Sigma}_0(\boldsymbol{\Phi}^\top \boldsymbol{\Phi}/n + \mu\boldsymbol{I}_p)^\dagger(\boldsymbol{\Phi}^\top \boldsymbol{y}/n)$$

$$= \overline{\mathrm{tr}}[\boldsymbol{W}^\top \boldsymbol{W}\boldsymbol{\Phi}(\tfrac{1}{n}\boldsymbol{\Phi}^\top \boldsymbol{W}^\top \boldsymbol{W}\boldsymbol{\Phi} + \lambda\boldsymbol{I}_p)^\dagger \boldsymbol{\Sigma}_0(\tfrac{1}{n}\boldsymbol{\Phi}^\top \boldsymbol{W}^\top \boldsymbol{W}\boldsymbol{\Phi} + \lambda\boldsymbol{I}_p)^\dagger \boldsymbol{\Phi}^\top \boldsymbol{W}^\top \boldsymbol{W}/n \cdot (\boldsymbol{y}\boldsymbol{y}^\top)]$$

$$\quad - \overline{\mathrm{tr}}[\boldsymbol{\Phi}(\boldsymbol{\Phi}^\top \boldsymbol{\Phi}/n + \mu\boldsymbol{I}_p)^\dagger \boldsymbol{\Sigma}_0(\boldsymbol{\Phi}^\top \boldsymbol{\Phi}/n + \mu\boldsymbol{I}_p)^\dagger \boldsymbol{\Phi}^\top/n \cdot (\boldsymbol{y}\boldsymbol{y}^\top)]$$

$$\simeq \overline{\mathrm{tr}}[(\boldsymbol{\Phi}\boldsymbol{\Phi}^\top/n + \mu\boldsymbol{I}_n)^\dagger(\boldsymbol{\Phi}\boldsymbol{\Sigma}_0\boldsymbol{\Phi}^\top/n + \mu'_{\boldsymbol{\Sigma}_0}\boldsymbol{I}_n)(\boldsymbol{\Phi}\boldsymbol{\Phi}^\top/n + \mu\boldsymbol{I}_n)^\dagger(\boldsymbol{y}\boldsymbol{y}^\top)]$$

$$\quad - \overline{\mathrm{tr}}[\boldsymbol{\Phi}(\boldsymbol{\Phi}^\top \boldsymbol{\Phi}/n + \mu\boldsymbol{I}_p)^\dagger \boldsymbol{\Sigma}_0(\boldsymbol{\Phi}^\top \boldsymbol{\Phi}/n + \mu\boldsymbol{I}_p)^\dagger \boldsymbol{\Phi}^\top/n \cdot (\boldsymbol{y}\boldsymbol{y}^\top)]$$

$$= \overline{\mathrm{tr}}[(\boldsymbol{\Phi}\boldsymbol{\Phi}^\top/n + \mu\boldsymbol{I}_n)^\dagger(\boldsymbol{\Phi}\boldsymbol{\Sigma}_0\boldsymbol{\Phi}^\top/n)(\boldsymbol{\Phi}\boldsymbol{\Phi}^\top/n + \mu\boldsymbol{I}_n)^\dagger(\boldsymbol{y}\boldsymbol{y}^\top)]$$

$$\quad + \mu'_{\boldsymbol{\Sigma}_0}\overline{\mathrm{tr}}[(\boldsymbol{\Phi}\boldsymbol{\Phi}^\top + \mu\boldsymbol{I}_n)^\dagger(\boldsymbol{y}\boldsymbol{y}^\top)(\boldsymbol{\Phi}\boldsymbol{\Phi}^\top + \mu\boldsymbol{I}_n)^\dagger]$$

$$\quad - \overline{\mathrm{tr}}[(\boldsymbol{\Phi}\boldsymbol{\Phi}^\top/n + \mu\boldsymbol{I}_n)^\dagger(\boldsymbol{\Phi}\boldsymbol{\Sigma}_0\boldsymbol{\Phi}^\top/n)(\boldsymbol{\Phi}\boldsymbol{\Phi}^\top/n + \mu\boldsymbol{I}_n)^\dagger(\boldsymbol{y}\boldsymbol{y}^\top)]$$

$$= \mu'_{\boldsymbol{\Sigma}_0}\overline{\mathrm{tr}}[(\boldsymbol{\Phi}\boldsymbol{\Phi}^\top + \mu\boldsymbol{I}_n)^\dagger(\boldsymbol{y}\boldsymbol{y}^\top)(\boldsymbol{\Phi}\boldsymbol{\Phi}^\top + \mu\boldsymbol{I}_n)^\dagger], \tag{36}$$

where in the third line, we used the second-order equivalence for freely weighted ridge resolvents from Lemma 22; in the fourth line, we employed the push-through identity multiple times. Substituting for $\mu'_{\boldsymbol{\Sigma}_0}$ from Lemma 22 in (36) and substituting this back into (35), we arrive at the desired decomposition. This completes the proof.

### C.2 Proof of Proposition 7

We use the path (4) with $k^*$ and $\mu^*$, and setting $\lambda^* = 0$:

$$\left(1 - \frac{\mathsf{df}(\widehat{\boldsymbol{\beta}}_{\boldsymbol{I},\mu^*})}{n}\right) = \left(1 - \frac{k^*}{n}\right).$$

This suggests that

$$k^* = \mathsf{df}(\widehat{\boldsymbol{\beta}}_{\boldsymbol{I},\mu^*}).$$

Note that $r := \mathrm{rank}(\boldsymbol{X}^\top \boldsymbol{X}) = \mathrm{rank}(\boldsymbol{G_I})$. By the definition of degrees of freedom, it follows that

$$\mathsf{df}(\widehat{\boldsymbol{\beta}}_{\boldsymbol{I},\mu^*}) = \mathrm{tr}[\boldsymbol{X}^\top \boldsymbol{X}(\boldsymbol{X}^\top \boldsymbol{X} + \mu^*\boldsymbol{I}_p)^\dagger]$$

$$= \sum_{i=1}^r \frac{s_i}{s_i + \mu^*} \le r = \mathrm{rank}(\boldsymbol{G_I}),$$

where $s_1, \ldots, s_r$ are non-zero eigenvalues of $\boldsymbol{X}^\top \boldsymbol{X}$. This finishes the proof.

### C.3 Technical lemmas

Recall from Appendix B.6 that we define $\lambda_0 = -\liminf_{n\to\infty} \lambda_{\min}^+(\boldsymbol{W}\boldsymbol{\Phi}\boldsymbol{\Phi}^\top \boldsymbol{W}^\top/n)$.

**Lemma 22** (General second-order equivalence for freely weighted ridge resolvents). Under the settings of Lemma 21, for any positive semidefinite $\boldsymbol{\Sigma}_0$ with uniformly bounded operator norm, for all $\lambda > \lambda_0$,

$$\tfrac{1}{n}\boldsymbol{W}^\top \boldsymbol{W}\boldsymbol{\Phi}(\tfrac{1}{n}\boldsymbol{\Phi}^\top \boldsymbol{W}^\top \boldsymbol{W}\boldsymbol{\Phi} + \lambda\boldsymbol{I}_p)^\dagger \boldsymbol{\Sigma}_0(\tfrac{1}{n}\boldsymbol{\Phi}^\top \boldsymbol{W}^\top \boldsymbol{W}\boldsymbol{\Phi} + \lambda\boldsymbol{I}_p)^\dagger \boldsymbol{\Phi}^\top \boldsymbol{W}^\top \boldsymbol{W}$$

$$\simeq (\tfrac{1}{n}\boldsymbol{\Phi\Phi}^\top + \mu \boldsymbol{I}_n)^\dagger (\tfrac{1}{n}\boldsymbol{\Phi\Sigma}_0\boldsymbol{\Phi}^\top + \mu'_{\boldsymbol{\Sigma}_0}\boldsymbol{I}_n)(\tfrac{1}{n}\boldsymbol{\Phi\Phi}^\top + \mu\boldsymbol{I}_n)^\dagger, \quad (37)$$

where $\mu'_{\boldsymbol{\Sigma}_0} \geq 0$ is given by:

$$\mu'_{\boldsymbol{\Sigma}_0} = -\frac{\partial\mu}{\partial\lambda}\lambda^2 \mathcal{S}'_{\boldsymbol{W}\boldsymbol{W}^\top}\big( - \tfrac{1}{n}\operatorname{tr}\big[\tfrac{1}{n}\boldsymbol{\Phi\Phi}^\top(\tfrac{1}{n}\boldsymbol{\Phi\Phi}^\top + \mu\boldsymbol{I}_n)^\dagger\big]\big)$$

$$\cdot \tfrac{1}{p}\operatorname{tr}\big[(\tfrac{1}{n}\boldsymbol{\Phi\Phi}^\top + \mu\boldsymbol{I}_n)^\dagger(\tfrac{1}{n}\boldsymbol{\Phi\Sigma}_0\boldsymbol{\Phi}^\top)(\tfrac{1}{n}\boldsymbol{\Phi\Phi}^\top + \mu\boldsymbol{I}_n)^\dagger\big]. \qquad (38)$$

*Proof.* We use the Woodbury matrix identity to write

$$\tfrac{1}{n}\boldsymbol{W}\boldsymbol{W}^\top\boldsymbol{\Phi}(\tfrac{1}{n}\boldsymbol{\Phi}^\top\boldsymbol{W}\boldsymbol{W}^\top\boldsymbol{\Phi} + \lambda\boldsymbol{I}_p)^\dagger\boldsymbol{\Sigma}_0(\tfrac{1}{n}\boldsymbol{\Phi}^\top\boldsymbol{W}\boldsymbol{W}^\top\boldsymbol{\Phi} + \lambda\boldsymbol{I}_p)^\dagger\boldsymbol{\Phi}^\top\boldsymbol{W}\boldsymbol{W}^\top$$
$$= \tfrac{1}{n}\boldsymbol{W}(\tfrac{1}{n}\boldsymbol{W}^\top\boldsymbol{\Phi\Phi}^\top\boldsymbol{W} + \lambda\boldsymbol{I}_m)^\dagger\boldsymbol{W}^\top\boldsymbol{\Phi\Sigma}_0\boldsymbol{\Phi}^\top\boldsymbol{W}(\tfrac{1}{n}\boldsymbol{W}^\top\boldsymbol{\Phi\Phi}^\top\boldsymbol{W} + \lambda\boldsymbol{I}_m)^\dagger\boldsymbol{W}^\top.$$

The equivalence in (37) and the inflation parameter in (38) now follow from the second-order result for feature sketch by substituting $\boldsymbol{W}$ for $\boldsymbol{S}$, $\boldsymbol{\Phi}$ for $\boldsymbol{\Phi}^\top$, and $\tfrac{1}{n}\boldsymbol{\Phi\Sigma}_0\boldsymbol{\Phi}^\top$ for $\boldsymbol{\Sigma}_0$ in (18). $\qquad\square$

# D Additional illustrations for Section 3

## D.1 Implicit regularization paths for bootstrapping

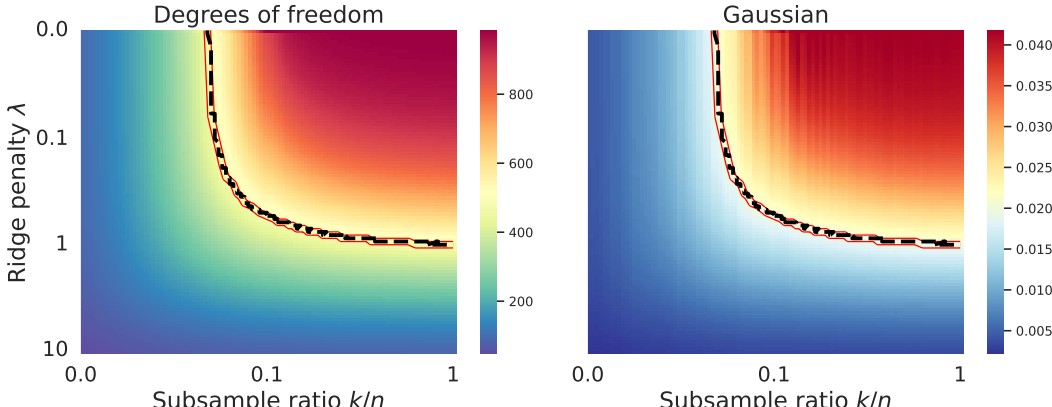

Figure 5: Equivalence under bootstrapping. The left panel shows the heatmap of degrees of freedom, and the right panel shows the random projection $\mathbb{E}_{\boldsymbol{W}}[\boldsymbol{a}^\top \widehat{\boldsymbol{\beta}}_{\boldsymbol{W},\lambda}]$ where $\boldsymbol{a} \sim \mathcal{N}(\boldsymbol{0}_p, \boldsymbol{I}_p/p)$. In both heatmaps, the red lines indicate the predicted paths using Equation (4), and the black dashed lines indicate the empirical paths obtained by matching empirical degrees of freedom. Despite the complexity of the theoretical path for bootstrapping, we observe that the empirical paths closely resemble it. Therefore, the theoretical path for sampling without replacement from (4) serves as a good approximation.

## D.2 Implicit regularization paths with non-uniform weights

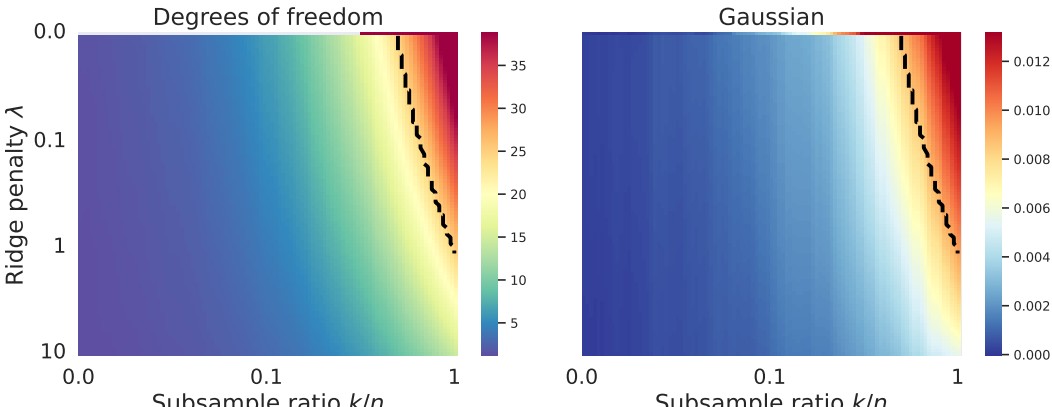

Figure 6: Equivalence under non-uniform weighting. The left panel shows the heatmap of degrees of freedom, and the right panel shows the random projection $\mathbb{E}_{\boldsymbol{W}}[\boldsymbol{a}^\top \widehat{\boldsymbol{\beta}}_{\boldsymbol{W},\lambda}]$, where $\boldsymbol{a} \sim \mathcal{N}(\boldsymbol{0}_p, \boldsymbol{I}_p/p)$. The weights $(\mathrm{diag}(\boldsymbol{W}))$ for observations are initially generated as $(9/10)^i$ for $i = 0, \ldots, n-1$, subsample $k$ entries from $\{1, \ldots, n\}$, zero out the other $n-k$ entries, and then normalized to have norm $k$. The black dashed lines indicate the empirical paths obtained by matching the empirical degrees of freedom.

# E  Additional illustrations for Section 4

## E.1  Rate illustration for ensemble risk against ensemble size

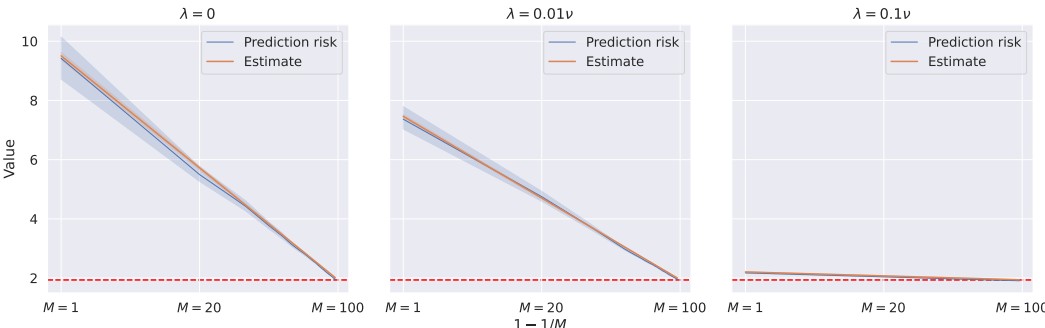

Figure 7: Risk equivalence for random feature structures when sampling without replacement. The solid lines represent the prediction risks and their estimates of the subsample ridge ensemble, and the red dashed lines indicate the prediction error of the full ridge predictor. The data and random features with the ReLU activation function are generated according to Appendix F.1 with $n = 5000$ and $p = 500$. The regularization level for the full ridge is set as $\mu = 1$, and each subsampled ridge ensemble is fitted with $M = 100$ randomly sampled subsampling matrices. For each value of $\lambda$, the subsample ratio is determined by solving Equation (4).

## E.2  Real data illustrations for implicit regularization paths

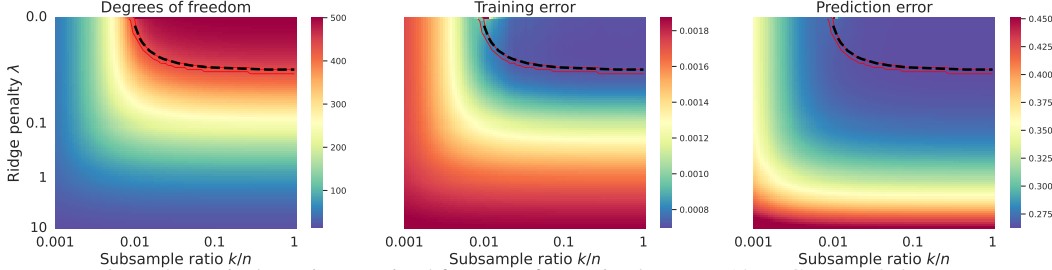

Figure 8: Equivalence in pretrained features of pretrained ResNet-18 on CIFAR-10 dataset.

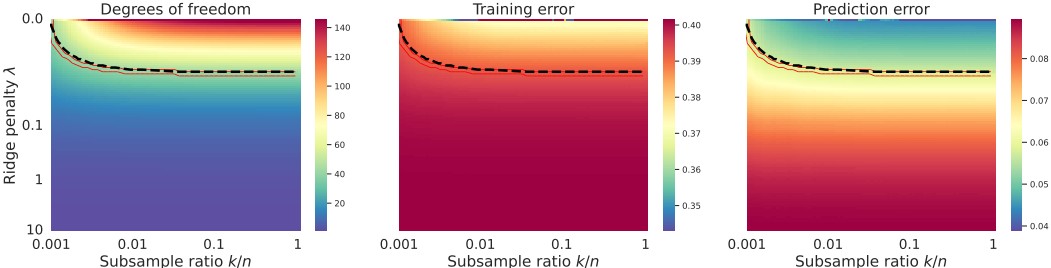

Figure 9: Equivalence in features of randomly initialized ResNet-18 on Fashion-MNIST dataset.

# F  Details of experiments

## F.1  Simulation details

The simulation settings are as follows.

- *Covariance model.* The covariance matrix of an auto-regressive process of order 1 (AR(1)) is given by $\mathbf{\Sigma}_{\mathrm{ar1}} \in \mathbb{R}^{d \times d}$, where $(\mathbf{\Sigma}_{\mathrm{ar1}})_{ij} = \rho_{\mathrm{ar1}}^{|i-j|}$ for some parameter $\rho_{\mathrm{ar1}} \in (0, 1)$. For the simulations, we set $\rho_{\mathrm{ar1}} = 0.25$.

- *Signal model.* Define $\boldsymbol{\beta}_0 = \frac{1}{5} \sum_{j=1}^{5} \boldsymbol{w}_{(j)}$ where $\boldsymbol{w}_{(j)}$ is the eigenvector of $\boldsymbol{\Sigma}_{\mathrm{ar1}}$ associated with the top $j$th eigenvalue $r_{(j)}$.

- *Response model.* We generated data $(\boldsymbol{x}_i, y_i) \in \mathbb{R}^d \times \mathbb{R}$ for $i = 1, \ldots, n$ from a nonlinear model:

$$y_i = \boldsymbol{x}_i^\top \boldsymbol{\beta}_0 + \frac{1}{p}(\|\boldsymbol{x}_i\|_2^2 - \mathrm{tr}[\boldsymbol{\Sigma}_{\mathrm{ar1}}]) + \varepsilon_i, \quad \boldsymbol{x}_i = \boldsymbol{\Sigma}_{\mathrm{ar1}}^{\frac{1}{2}} \boldsymbol{z}_i, \quad z_{ij} \overset{iid}{\sim} \frac{t_5}{\sigma_5}, \quad \varepsilon_i \sim \frac{t_5}{\sigma_5},$$

$$\text{(M-AR1)}$$

where $\sigma_5 = \sqrt{5/3}$ is the standard deviation of $t_5$ distribution.

The benefit of using the above nonlinear model is that we can clearly separate the linear and the nonlinear components and compute the quantities of interest because $\boldsymbol{\beta}_0$ happens to be the best linear projection.

The linear, random, and kernel features are generated as follows.

- *Linear features.* For a given feature dimension $p$, we use $d = p$ raw features from (M-AR1) as linear features.

- *Random features.* For generating random features, we use $d = 2p$ raw features from (M-AR1) and sample a randomly initialized weight matrix $\boldsymbol{F} \in \mathbb{R}^{p \times d}$ whose entries are i.i.d. samples from $\mathcal{N}(0, d^{-1/2})$. Then the transform feature is given by $\widetilde{\boldsymbol{x}}_i = \varphi(\boldsymbol{F}\boldsymbol{x}_i) \in \mathbb{R}^p$, where $\varphi$ is a nonlinear transformation and set to be ReLU function in our experiment.

- *Kernel features.* For kernel features, we use $d = p$ raw features from (M-AR1) to construct the kernel matrix.

In the simulations, the estimates are averaged across 20 simulations with different random seeds.

### F.2   Experimental details in Section 4.3

Following the similar experimental setup in [20], we use residual networks to extract features on several computer vision datasets, both at random initialization and after pretraining. More specifically, we consider ResNet-{18, 34, 50, 101} applied to the CIFAR-{10,100} [9], Fashion-MNIST [21], Flowers-102 [14], and Food-101 [4] datasets. All random initialization was done following [8]; pretrained networks (obtained from PyTorch) were pretrained on ImageNet, and the outputs of the last pretrained layer on each dataset mentioned above were used as the embedding feature $\boldsymbol{\Phi}$.

After obtaining the embedding features from the last layer of the neural network model, we further normalize each row of the pretrained feature to have a norm of $p$, and center the one-hot labels to have zero means. To reduce the computational burden, we only consider the first 10 one-hot labels of all datasets. For datasets with different data aspect ratios, we stratify 10% of the training samples as the training set for the CIFAR-100 dataset. The training and predicting errors are the mean square errors on the training and test sets, respectively, aggregated over all the labels.

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

Table 2: Summary of pretrained features from different real datasets.

| Dataset | Model | Number of train samples | Number of test samples | Number of pretrained features |
|---|---|---|---|---|
| Fashion-MNIST | ResNet-18 init. | 60000 | 10000 | 512 |
| CIFAR-10 | ResNet-18 pretr. | 50000 | 10000 | 512 |
| CIFAR-100 (subset) | ResNet-50 pretr. | 5000 | 10000 | 2048 |
| Flowers-102 | ResNet-50 pretr. | 2040 | 6149 | 2048 |
| Food-101 | ResNet-101 pretr. | 75750 | 25250 | 2048 |

[5] Dobriban, E. and Sheng, Y. (2020). Wonder: Weighted one-shot distributed ridge regression in high dimensions. *Journal of Machine Learning Research*, 21(66):1–52.

[6] Dobriban, E. and Sheng, Y. (2021). Distributed linear regression by averaging. *The Annals of Statistics*, 49(2):918–943.

[7] Hastie, T., Montanari, A., Rosset, S., and Tibshirani, R. J. (2022). Surprises in high-dimensional ridgeless least squares interpolation. *The Annals of Statistics*, 50(2):949–986.

[8] He, K., Zhang, X., Ren, S., and Sun, J. (2015). Delving deep into rectifiers: Surpassing human-level performance on imagenet classification. In *International Conference on Computer Vision*.

[9] Krizhevsky, A. and Hinton, G. (2009). Learning multiple layers of features from tiny images.

[10] Lee, D., Moniri, B., Huang, X., Dobriban, E., and Hassani, H. (2023). Demystifying disagreement-on-the-line in high dimensions. In *International Conference on Machine Learning*.

[11] LeJeune, D., Patil, P., Javadi, H., Baraniuk, R. G., and Tibshirani, R. J. (2024). Asymptotics of the sketched pseudoinverse. *SIAM Journal on Mathematics of Data Science*, 6(1):199–225.

[12] Mel, G. and Pennington, J. (2021). Anisotropic random feature regression in high dimensions. In *International Conference on Learning Representations*.

[13] Mingo, J. A. and Speicher, R. (2017). *Free Probability and Random Matrices*, volume 35. Springer.

[14] Nilsback, M.-E. and Zisserman, A. (2008). Automated flower classification over a large number of classes. In *Indian Conference on Computer Vision, Graphics and Image Processing*.

[15] Patil, P. and Du, J.-H. (2023). Generalized equivalences between subsampling and ridge regularization. *Advances in Neural Information Processing Systems*.

[16] Patil, P., Du, J.-H., and Kuchibhotla, A. K. (2023). Bagging in overparameterized learning: Risk characterization and risk monotonization. *Journal of Machine Learning Research*, 24(319):1–113.

[17] Patil, P. and LeJeune, D. (2024). Asymptotically free sketched ridge ensembles: Risks, cross-validation, and tuning. In *International Conference on Learning Representations*.

[18] Sahraee-Ardakan, M., Emami, M., Pandit, P., Rangan, S., and Fletcher, A. K. (2022). Kernel methods and multi-layer perceptrons learn linear models in high dimensions. *arXiv preprint arXiv:2201.08082*.

[19] Voiculescu, D. V. (1997). *Free Probability Theory*. American Mathematical Society.

[20] Wei, A., Hu, W., and Steinhardt, J. (2022). More than a toy: Random matrix models predict how real-world neural representations generalize. In *International Conference on Machine Learning*.

[21] Xiao, H., Rasul, K., and Vollgraf, R. (2017). Fashion-mnist: a novel image dataset for benchmarking machine learning algorithms. *arXiv preprint arXiv:1708.07747*.

