# OpenReview forum: "Implicit Regularization Paths of Weighted Neural Representations"
_NeurIPS.cc/2024/Conference — NeurIPS 2024 poster_

### Official Review · Reviewer_HnNV · 2024-07-09

**Soundness:** 3
**Presentation:** 3
**Contribution:** 4
**Rating:** 7
**Confidence:** 2

**Summary:**

Neural networks are a powerful tool for extracting features from data, but these features can be very high-dimensional. This high dimensionality can be a bottleneck for training machine learning models, requiring computational power and memory. This manuscript investigates implicit regularization through subsampling and draws a connection to weighted regression problems. The authors provide theoretical results for the claimed connection between implicit regularization via subsampling and regression. The authors provide numerical evidence of their investigations.

**Strengths:**

The manuscript is well-written and provides an interesting insight into the connection between subsampling approaches and explicit regularization.

**Weaknesses:**

Some works have provided between subsampling and iterative least squares approaches, see Slagel et al., Sampled Tikhonov Regularization for Large Linear Inverse Problems.

**Questions:**

see weaknesses.

**Limitations:**

yes.

---

> ### Author Rebuttal · Authors · 2024-08-05
>
> We sincerely thank the reviewer for their time and valuable feedback and acknowledging the strengths of our paper!
> Thanks also for a nice reference pointer!
> Below, we expand more on this reference and connections to our work.
>
> **Weakness** and **Question**
>
> - **[W1] and [Q1] Related work on iterative least squares.**
> We thank the reviewer for pointing out this great relevant reference about the relationship between subsampling and iterative least squares approaches. We will definitely add the missing reference to the related work in our revision.
> Compared to the work by [SCC+19], there are several differences (and some improvements) that our work offers:
>   - *Problem setup.* [SCC+19] show the equivalence between the iterates along the (randomly) subsampled iterative methods and full Tikhonov regularized estimator (also known as generalized ridge regression estimator). On the other hand, we show equivalence between the subsampled (and ensemble) weighted ridge estimators and ridge(less) estimators on full data (at a different regularization). It is definitely related but these are somewhat complementary directions.
>   - *Regime of interest.* [SCC+19] implicitly only analyze an underparameterized regime (where the sample size $n$ is required to be larger than the feature dimension $p$). This is because the data matrix is required to have full column rank, which will only happen in the underparameterized regime. Moreover, they only consider the target ridge estimator with $\lambda > 0$. In comparison, our paper considers both underparameterized and overparameterized regimes (allowing for both $n > p$ and $n < p$). Furthermore, we also allow the target ridge with $\lambda = 0$ (also known as ridgeless) and potentially negative $\lambda$ as well.
>   - *Sampling strategies.* [SCC+19] only consider random subsampling matrices, which are akin to orthogonal matrices (in expectation). In comparison, our work allows for general weighting matrices that we assume to be asymptotically free from the data matrices.
>   - *Data model.* [SCC+19] assume a well-specified linear model. Our work does not assume any specific model for the response and only require finite average response energy (in the limit).
>   - *Types of equivalences.* [SCC+19] show the equivalence of the estimators in expectation (over the randomness in subsampling). On the other hand, we show asymptotic equivalences for both the estimators and quadratic risks that hold almost surely (over the randomness in weight and data matrices). In addition, we also establish entire (data-dependent) paths of equivalences for the weighted ensembled ridge estimators.
>
> **Reference**
> - [SCC+19] Slagel J. T., Chung, J., Chung M., Kozak D., and Tenorio L. Sampled Tikhonov regularization for large linear inverse problems. Inverse Problems, 2019.

---

> > ### Comment · Reviewer_HnNV · 2024-08-12
> >
> > Thank you for your response. The authors have addressed some of my concerns and I have slightly increased my rating.

---

> > > ### Author Response · Authors · 2024-08-13
> > >
> > > Thanks, we are happy that our response addressed your concerns! Thanks again for the reference pointer! We will definitely mention this and the comparison in our revision.

---

### Official Review · Reviewer_nM8u · 2024-07-12

**Soundness:** 3
**Presentation:** 3
**Contribution:** 3
**Rating:** 7
**Confidence:** 2

**Summary:**

This paper studies the weighted linear regression problem where the feature matrix is left-multiplied with a random matrix $\mathbf{W}$ denoting the sample weighting. Under the assumption of the asymptotic freeness between this weighting matrix and the feature matrix, the paper shows that the weighted ridge regression estimator is equivalent to a simple ridge regression estimator in the limit of infinite samples by establishing a connection between the regularization strengths of the weighted and the simple ridge regression. Based on this regularization path, the paper gives illustrative examples of concrete weighting matrix and feature matrix. Moreover, the paper also shows a equivalence between the risk of ensemble weighted and simple ridge regression under the same regularization path. Based on this result, the paper derives an optimal sub-sample size for the ensemble training. Theoretical results of this paper is accompanied by experimental verification.

**Strengths:**

1. The paper considers a general scheme where $\mathbf{W}$ can be any random weighting matrix, which can generalize beyond the sub-sampling scenario. The assumption on the asymptotic freeness between the weighting matrix and the feature matrix is quite relaxed.
2. The paper derives an exact relationship between the regularization of the weighted ridge regression and that of the simple ridge regression in the limit of $n\rightarrow\infty$. This relationship is verified by the experiments presented in the paper.
3. The paper also derives the equivalence between risks of the ensemble weighted ridge regression and the simple ridge regression. The equivalence demonstrates a bias-variance trade-off and also validate the benefit of using a larger ensemble size.
4. Although the results is based on sophisticated mathematical notions, the paper uses intuitive explanation to make the results more comprehensible by a broader audience.

**Weaknesses:**

1. The results derived in the paper has limited applicability. In particular, although it would be interesting to know the equivalence between the weighted ridge regression and the simple ridge regression, the regularization path does not tell much about how should we choose a weighting matrix, even assuming the optimal ridge regression regularization is known. Moreover, the optimal sub-sample size in Proposition 7 involves computing $\mu^*$ which cannot be done easily. Although the paper discussed a way to measure $\mu^*$, it is still computationally heavy, and even in the case where $\mu^*$ is known, one might not be able to measure the degree of freedom easily.
2. The paper considers an asymptotic regime where the sample size goes to infinity. The results in the paper will have better implications if a dependency on $n$ is given.
3. The paper's result is derived under the ridge regression setting. Although one can always choose $\Phi$ to be the output of some pre-trained neural network, the results barely connect to any property of neural network training.

**Questions:**

1. Are there any example in real-world applications where the weighting matrix is not a diagonal matrix?
2. Based on the theoretical results in the paper, what would be the benefit of using a sub-sampling matrix with non-binary diagonal entries?

**Limitations:**

The paper has a good discussion of its limitations.

---

> ### Author Rebuttal · Authors · 2024-08-05
>
> Thank you for taking the time to review our paper and providing valuable suggestions!
> Thank you also for the nice questions!
> We appreciate all the feedback and have addressed the weaknesses and questions below.
>
> **Weaknesses**
>
> - **[W1] Practical applicability.** Both Theorems 1 and 2 provide "data-dependent" paths through path (2) and path (4), respectively. By fixing target regularization $\mu$, one can compute the degrees of freedom of the full ridge estimator using training data (see line 149). This allows us to numerically solve for the path since the degrees of freedom of the weighted ridge estimator can also be computed using the same formula.
> Statisticians have two hyperparameter choices: the type of weighting and the regularization level. We offer a method to tune these in Section 4.2, specifically for subsampling-based weighting, which can be generalized to other parameterized weighting schemes. The oracle optimal subsample size in Proposition 7 may include unknown parameters like $\mu^*$, but for practical purposes, knowing these parameters is unnecessary.
> The equivalence implies one can fix a small regularization level and adjust the subsample size. The method in Section 4.2 leverages this, providing a data-driven way to tune the optimal subsample size. Although it requires fitting multiple models, these can be computed independently and distributed, reducing computational time. In Section 4.3, we apply this approach to real-world datasets (e.g., Figure 4) to predict explicit regularization matching the implicit regularization from subsampling.
> When $\mu^*$ is known, the situation simplifies. We can compute the degrees of freedom of the full ridge estimator at $\mu^*$ and use the data-dependent path to find various weights and regularization levels that yield the optimal estimator.
>
> - **[W2] Asymptotic analysis.** We examine proportional asymptotic regimes where the sample size $n$, feature dimension $p$, and subsample size $k$ all tend to infinity, but their ratios $p/n$ and $p/k$ converge to constants. In this analysis, both data and weighting matrices are indexed by $n$, a common approach in random matrix theory. This asymptotic method simplifies proofs and highlights essential problem characteristics under minimal assumptions.
> Extending our results to finite samples is possible but requires additional assumptions, as precise error bounds depend on the specifics of feature and response distributions. For subsampling weights, techniques from [KY17], [L22], [CM22], and [HX23] may yield non-asymptotic versions of some statements in our paper. However, for general weights satisfying the asymptotic freeness assumption, obtaining non-asymptotic results is challenging. This remains an active area of research in free probability theory.
>
> - **[W3] Relation to neural network training.** Our results are indifferent to the type of neural network training that proves advantageous. Other theoretical analyses link the generalization error of trained models to network properties and data distribution parameters (e.g., [JGH18], [AP20], [BMR21], [MM22]). For example, in random features regression, similar to a two-layer network with random first-layer weights, the generalization error depends on the distributional properties of the weights and the activation function. Although these analyses offer insights into how network properties affect generalization error, the risk expressions become complicated with additional layers.
> Our work is different in style from these works. In particular, we do not analyze the risk of either the full model or the subsampled models in isolation. Instead, we relate these two sets of models, allowing us to maintain weak assumptions about the features. Note that it is possible to combine our equivalence results with the aforementioned line of work and this gives a way to port the insights from one model to other equivalent models.
>
> **Questions**
>
> Sincere thanks for the great questions!
>
> - **[Q1] Non-diagonal weighting matrix.**  Observation sketching involves taking random linear combinations of the rows of the data matrix, resulting in a non-diagonal weighting matrix. This technique can be useful for privacy, scrambling identifiable information, or mitigating the effects of non-i.i.d. data in time series or spatial data. We will discuss the non-diagonal weighting matrix in our revision.
>
> - **[Q2] Non-binary diagonal weighting matrix.** Even with subsampling, a non-binary diagonal weighting matrix is possible. For instance, sampling with replacement or a specific distribution results in non-binary diagonal weighting matrices, as illustrated in Figures 6 and 7 of the supplement. Other examples include inverse-variance weighting to address heterogeneous variations when responses have different variances for different units.
>
> **References**
>
> - [KY17] Antti Knowles and Jun Yin. Anisotropic local laws for random matrices. Probability Theory and Related Fields, 169:257–352, 2017
> - [JGH18] Jacot, A., Gabriel, F., and Hongler, C. Neural tanget kernel: Convergence and generalization in neural networks. In NeurIPS, 2018.
> - [AP20] Adlam, B.~ and Pennington, J. The neural tangent kernel in high dimensions: Triple descent and a multi-scale theory of generalization. In ICML, 2020.
> - [BMR21] Bartlett, P., Montanari, A., and Rakhlin, A. Deep learning: A statistical viewpoint. Acta Numerica, 2021.
> - [L22] Cosme Louart. Sharp bounds for the concentration of the resolvent in convex concentration settings. arXiv:2201.00284, 2022
> - [CM22] Chen Cheng and Andrea Montanari. Dimension free ridge regression. arXiv:2210.08571, 2022.
> - [MM22] Mei, S. and Montanari, A. The generalization error of random features regression: Precise asymptotics and the double descent curve. In Communications on Pure and Applied Mathematics, 2022.
> - [HX23] Han, Qiyang, and Xiaocong Xu. The distribution of Ridgeless least squares interpolators. arXiv preprint arXiv:2307.02044, 2023.

---

> > ### Comment · Reviewer_nM8u · 2024-08-08
> > **Response to the Author's Rebuttal**
> >
> > Thank you for your response. The authors response to [W1] and [Q1] further convinced me about the applicability of the results. Moreover, the author also discussed how to extend the result to finite $n$ in the response to [W2]. Therefore, I have raised my score accordingly.

---

> > > ### Author Response · Authors · 2024-08-13
> > >
> > > Great, thanks! We are happy that you found our response helpful and convincing! Thanks again for all the feedback, which we will definitely use in our revision.

---

### Official Review · Reviewer_cp5Y · 2024-07-13

**Soundness:** 2
**Presentation:** 1
**Contribution:** 3
**Rating:** 5
**Confidence:** 3

**Summary:**

This paper investigates the implicit regularization effects of weighted pretrained features. It establishes a path of equivalence between different weighting matrices and ridge regularization with matching effective degrees of freedom. The study extends results to structured features and ensembles, providing theoretical validation for random and kernel features, and proposes an efficient cross-validation method to finetune subsampled pretrained representations in practice.

**Strengths:**

- Table 1 provides a clear overview of the previous works, along with the contributions and organization of this paper.
- The theoretical results are comprehensive and strong in terms of relaxed assumptions on the features and weights.
- The theoretical results bring insights into designing a practical cross-validation algorithm, whose effectiveness is supported by experiments.

**Weaknesses:**

- While Table 1 and the summary of results provide a relatively clear view of the theoretical contributions, the abstract and the motivation part of the introduction read somewhat confusing. For example, the "path of equivalence" is repeatedly mentioned in the abstract and introduction as one of the main contributions, but without a clear explanation/intuition of what it means until later.
- The problem setup and notations are not clearly stated before being used. For example, in terms of setup, in Assumption A, it's unclear what the convergence refers to in "$W^\top W$ and $\Phi \Phi^\top/n$ converge almost surely to bounded operators". In terms of notations, $\|\cdot\|_{tr}$ and $\overline{tr}(\cdot)$ are used without clear definitions for reference.
- Overall, the layout of the results is a little bit hard to follow, partially due to the lack of clear problem setup and notations mentioned before, and partially due to presuming prior knowledge of specific technical tools like free probability theory.

**Questions:**

- In Definition A, what's the precise definition of $\overline{tr}(\cdot)$? What does it mean by saying "$W^\top W$ and $\Phi \Phi^\top/n$ converge almost surely to bounded operators"? If it refers to the convergence as $n \to \infty$, then does it mean the analysis is conducted in the asymptotic regime?
- In line 196, if (5) implies $\lambda < \mu$, isn't the regularization level $\lambda$ of the subsampled estimator lower (instead of higher) than that of the full estimator $\mu$?
- It's somehow counter-intuitive that the equivalence path results depend only on the subsample fraction $k/n$ under the "general" data assumption. Common analyses in the asymptotic regime reduce data properties to the subsample ratio $k/n$ usually based on strong assumptions on the data distribution like Gaussian, while data-agnostic subsampling on adversarial data should bring inevitable compromise compared to full data (e.g., with a direction that is only learnable through a single data point). It seems that the "infinitesimally freeness" in Assumption A implicitly circumvents this issue, but it's unclear in the current discussion. For example:
    - Whether Assumption A enforces a "nice" data distribution, a "good" subsampling strategy, or both?
    - If Assumption A implies a "good" subsampling strategy, can such subsampling matrix $W$ be efficiently constructed in practice?

**Limitations:**

Limitations are well-discussed in the Conclusion. Some further limitations are mentioned in Weaknesses and Questions.

---

> ### Author Rebuttal · Authors · 2024-08-05
>
> Thank you for the encouragement and comments! While we appreciate all the feedback, we believe that the main concerns raised are related to the clarity of exposition and can be easily addressed easily. Below, we address all the weaknesses and questions on a point-by-point basis.
> We respectfully request that the reviewer reconsider the scores based on the technical contributions of the paper, which we believe are significant.
>
> **Weaknesses**
>
> - **[W1] Explanation/intuition of main contributions.**
> Path of equivalence refers to data-dependent set of weighted ridge estimators $(\mathbf{W} , \lambda)$ that connect to the unweighted ridge estimator $(\mathbf{I}, \mu)$ is defined in terms of "matching" effective degrees of freedom of component estimators in the set (see lines 161-163). This path allows us to relate the performance and properties of models fitted on full datasets to those fitted on subsampled datasets.
> In the Abstract (lines 7-9), we refer to it as "a path of equivalence connecting different weighting matrices and ridge regularization with matching effective degrees of freedom." This is to keep the abstract within the recommended 4-6 lines. In the introduction (lines 39-41), we explain mathematically what a path means. Based on your feedback, we will add a line in the Abstract to even more clearly state what a path of equivalence means.
> - **[W2] Problem setup.**
> In terms of problem setup, we explain the meaning of ``converge almost surely to bounded operators'' below. In free probability theory and random matrix theory, matrices as linear transformations are viewed as operators. A bounded operator is a linear transformation on a vector space where the magnitude of the output vector is bounded by a constant multiple of the magnitude of the input vector. In matrix terms, this means that the eigenvalues of the matrix are bounded. For us, this means that the maximum eigenvalues of the matrices $\mathbf{W}^\top \mathbf{W}$ and $\mathbf{\Phi} \mathbf{\Phi}^\top / n$ are almost surely bounded in the limit as $n \to \infty$. This is a standard assumption in free probability theory and keeps the estimators and their risks bounded in the limit.
> - **[W2] Notational clarity.**
> All the important notation in the paper is well-defined and exhaustive. We have an entire subsection of the Appendix on this (lines 552-580). Not all of this notation is in the main paper because of the page limit of 9 pages due to the space limit, but note that the meaning of $\overline{tr}(\cdot)$ is explained in the main paper (130-131). The notation of $tr(\cdot)$, $\overline{tr}(\cdot)$, and $\|\cdot\|_{tr}$ is defined on lines 572-573, 577. This notation is standard in the fields of random matrix theory and free probability theory.
> While we understand that not all readers will be familiar with these fields, it is not feasible for us to include every notation and definition in the main paper due to page limits, and hence, we added this in the Appendix. Based on your feedback, we will include a paragraph on notation in the main paper in our revision.
> - **[W3] Layout and accessibility of results.**
> The Summary of Results (line 55-77) outlines the paper's contents: Section 2 covers preliminaries, Section 3 discusses implicit regularization paths and examples of weight and feature matrices, and Section 4 addresses prediction risk asymptotics, risk estimation, optimal tuning, and validation on real-world datasets. Section 5 concludes with limitations and outlook. We will add a separate organization section in the revised version.
> To aid readers unfamiliar with random matrix and free probability theories, Appendix A provides a self-contained background, including:
>   - Appendix A.1: Basics of free probability theory.
>   - Appendix A.2: Useful transforms and their relationships.
>   - Appendix A.3: Asymptotic ridge resolvents.
>
> Advanced tools are necessary for such general results, and we try to make them accessible with Appendix A. We welcome additional suggestions from reviewers but we consider the use of tools from other fields in machine learning as a strength rather than a weakness.
>
> **Questions**
>
> - **[Q1] Definitions.** The definition of $\overline{tr}$ is given in the notation paragraph (line 577) of the Appendix. See also our response to [W2]. The analysis in the current paper is indeed conducted in the asymptotic regime, in which the sequence of random matrices (with possibly growing dimensions) is indexed by $n$ (see lines 151-156).
> - **[Q2] Comparing regularization strength of subsample and full estimators.**
> We thank the reviewer for pointing out the typo in wording. Indeed, the regularization level $\lambda$ of the subsampled estimator is lower than that of the full estimator $\mu$. We will correct this in the revised version.
> - **[Q3] Equivalence path results.** Thanks, this is a great question!
> Common analyses in the asymptotic regime often rely on strong assumptions like Gaussian distribution, while our approach is more general. A key result is that equivalence paths are characterized by degrees of freedom.
>   - Assumption A does not require a "nice" *marginal* data and weighting distribution. But it relies on the "infinitesimal freeness" of the (data, weight) pair. Intuitively, for subsampling, it means that the pair needs to be "sufficiently" independent (in the limit). In particular, adversarial subsampling may not be permitted as such.
>   -  Yes. *Independent* random subsampling is easy to construct. In Section 3.2, we construct various examples that satisfy Assumption A. In general, it is also empirically satisfied on real data (as in Figure 3 and Appendix 3.2).
>
> Given the significant technical contributions and the ease of addressing the reviewer's concerns, we kindly request the reviewer to reconsider the score. We will improve the presentation of the paper based on the feedback. We believe the technical contributions and practical relevance of our work merit a more favorable evaluation.

---

> > ### Comment · Reviewer_cp5Y · 2024-08-12
> >
> > Thanks a lot for the detailed response. Assuming a better presentation, I will increase my score to 5.

---

> > > ### Author Response · Authors · 2024-08-13
> > >
> > > Thanks, we appreciate it! And yes, we will definitely improve on the presentation as indicated in our response above.

---

### Author Rebuttal · Authors · 2024-08-06

We sincerely thank all the reviewers for taking the time to review our paper. We appreciate the constructive comments and valuable feedback.

All three reviewers have acknowledged several key strengths of our paper.

- Reviewer **cp5Y** liked the clear overview of previous works provided in Table 1, the comprehensive and strong theoretical results with relaxed assumptions, and the practical insights from our theory that lead to an effective cross-validation algorithm.
- Reviewer **nM8u** highlighted the generality of our results, the relaxed assumptions on asymptotic freeness that our results require, the exact relationship established between the implicit regularization effects, and the comprehensible presentation despite sophisticated mathematical theory.
- Reviewer **HnNV** appreciated the well-written manuscript, the interesting insights into the connection between subsampling approaches and explicit regularization, and the theoretical (for a variety of standard feature models) and numerical (on a variety of datasets) evidence supporting our investigations.

The reviewers have also pointed out potential weaknesses and places for improvements in the paper.

- Reviewer **cp5Y** mentioned some confusion regarding the notation, terminology, and layout of the paper.
- Reviewer **nM8u** noted the focus on the asymptotic regime.
- Reviewer **HnNV** suggested additional related work on subsampling and iterative least squares approaches.

We have carefully addressed each reviewer's questions and points of weakness in a point-by-point manner below. To improve readability and for space reasons, we have divided our responses into different posts. We apologize for the length of our response, but we provide thorough and comprehensive answers to all the reviewer's questions and concerns.

In particular, we have tried to clear some misunderstandings of Reviewer **cp5Y** with regard to notation, terminology, and organization. We briefly describe this below for reviewers' convenience.

- **Notation**: The reviewer may have missed this but we actually have a dedicated notation section in the Appendix (lines 552-580) that summarizes all the key notations used in the paper. Some of the important notation (such as average trace) is also mentioned in the main paper where it is used (lines 130-131). We understand that the paper is notation-heavy, and the notation may seem foreign for readers not familiar with random matrix theory and free probability theory, but the notation we use is standard in these fields.

- **Terminology**: We understand that some of the terminology in the paper may be specific to the literature in this area. But note that we define the term "paths of equivalence" in the Introduction (lines 38-41) and intuitively explain this in the Abstract (lines 7-9). We will try to explain this in even more detail in our revision.

- **Organization**: With regards to the layout, we have a section-wise summary of results in the paper in the Summary of Results paragraph (lines 55-77). We will add an explicit organization section in our revision (with the additional page provided in the final version).

We have kindly asked Reviewer **cp5Y** to reconsider the score given after clearing these misunderstandings.

We believe that our response comprehensively addresses the concerns raised by the reviewers. We are open to any further feedback and look forward to subsequent comments from the reviewers.

---

### Decision · Program_Chairs · 2024-09-25

**Decision:**

Accept (poster)

**Comment:**

After the rebuttal, all reviewers gave positive scores. The AC recommends acceptance based on the following merits:

* The submission presents strong theoretical results based on relaxed assumptions on both the weighting matrix and the feature matrix. It demonstrates the precise relationship between the regularization of weighted ridge regression and simple ridge regression.

* The proposed theory introduces a novel approach to tuning regularization strengths through subsampling. The experiments in Figure 4 confirm the theoretical relationship between risks and validate the effectiveness of the tuning procedure.

The authors are encouraged to improve the organization of the paper following the advice of reviewer cp5Y in the final version.